# Finite-size security of continuous-variable quantum key distribution with digital signal processing

Takaya Matsuura [1], Kento Maeda[1], Toshihiko Sasaki [1,2] & Masato Koashi [1,2 ✉]

In comparison to conventional discrete-variable (DV) quantum key distribution (QKD), continuous-variable (CV) QKD with homodyne/heterodyne measurements has distinct advantages of lower-cost implementation and affinity to wavelength division multiplexing. On the other hand, its continuous nature makes it harder to accommodate to practical signal processing, which is always discretized, leading to lack of complete security proofs so far. Here we propose a tight and robust method of estimating fidelity of an optical pulse to a coherent state via heterodyne measurements. We then construct a binary phase modulated CV-QKD protocol and prove its security in the finite-key-size regime against general coherent attacks, based on proof techniques of DV QKD. Such a complete security proof is indispensable for exploiting the benefits of CV QKD.

---

[1] Department of Applied Physics, Graduate School of Engineering, The University of Tokyo, 7-3-1 Hongo, Bunkyo-ku, Tokyo 113-8656, Japan. [2] Photon Science Center, Graduate School of Engineering, The University of Tokyo, 7-3-1 Hongo, Bunkyo-ku, Tokyo 113-8656, Japan. ✉email: koashi@qi.t.u-tokyo.ac.jp

 1

Quantum key distribution (QKD) aims at generating a secret key shared between two remote legitimate parties with information-theoretic security, which provides secure communication against an adversary with arbitrary computational power and hardware technology. Since the first proposal in 1984[1], various QKD protocols have been proposed with many kinds of encoding and decoding schemes. These protocols are typically classified into two categories depending on the detection methods. One of them is called discrete-variable (DV) QKD, which uses photon detectors and includes earlier protocols such as BB84[1] and B92[2] protocols. The other is called continuous-variable (CV) QKD, which uses homodyne and heterodyne measurements with photo detectors[3–5]. See refs. [6,7] for comprehensive reviews of the topic.

Although DV QKD is more mature and achieves a longer distance if photon detectors with low dark count rates are available, CV QKD has its own distinct advantages for a short distance. It can be implemented with components common to coherent optical communication technology and is expected to be cost-effective. Excellent spectral filtering capability inherent in homodyne/heterodyne measurements suppresses crosstalk in wavelength division multiplexing (WDM) channels. This allows multiplexing of hundreds of QKD channels into a single optical fiber[8] as well as co-propagation with classical data channels[9–15], which makes integration into existing communication network easier.

One major obstacle in putting CV QKD to practical use is the gap between the employed continuous variables and mandatory digital signal processing. The CV-QKD protocols are divided into two branches depending on whether the modulation method of the encoder is also continuous, or it is discrete. The continuous modulation protocols usually adopts Gaussian modulation, in which the sender chooses the complex amplitude of a coherent-state pulse according to a Gaussian distribution[3–5,16,17] (see refs. [18,19] for a review). This allows powerful theoretical tools such as Gaussian optimality[20,21], and complete security proofs for a finite-size key and against general attacks have been given[22]. To implement Gaussian protocols with a digital random-number generator and digital signal processing, it is necessary to approximate the continuous distribution with a constellation composed of a large but finite number of complex amplitudes[23,24]. This is where difficulty arises, and the security analysis has been confined to the asymptotic regime and collective attacks. The other branch gives priority to simplicity of the modulation and uses a very small (usually two to four) number of amplitudes[25–28]. As for the security analysis, the status is more or less similar to the Gaussian constellation case, and current security proofs are either in the asymptotic regime against collective attacks[29–32] or in the finite-size regime but against more restrictive attacks[33,34]. Hence, regardless of approaches, a complete security proof of CV QKD in the finite-size regime against general attacks has been a crucial step yet to be achieved.

Here we achieve the above step by proposing a binary phase-modulated CV-QKD protocol with a complete security proof in the finite-size regime against general attacks. The key ingredient is an estimation method using heterodyne measurement developed in this paper, which is suited for analysis of confidence region in the finite-size regime. The outcome of heterodyne measurement, which is unbounded, is converted to a bounded value by a smooth function such that its expectation is proved to be no larger than the fidelity of the input pulse to a coherent state. This allows us to use a standard technique to derive a lower bound on the fidelity with a required confidence level in the finite-size regime. The fidelity as a measure of disturbance in the binary modulated protocol is essentially the same as what is monitored through bit errors in the B92 protocol[2,35,36]. This allows us to construct a security proof based on a reduction to

distillation of entangled qubit pairs[37,38], which is a technique frequently used for DV-QKD protocols.

## Results

**Estimation of fidelity to a coherent state.** We first introduce a test scheme to estimate the fidelity between an optical state $\rho$ and the vacuum state $|0\rangle \langle 0|$ through a heterodyne measurement. For a state $\rho$ of a single optical mode, the heterodyne measurement produces an outcome $\hat{\omega} \in \mathbb{C}$ with a probability density

$$q_\rho(\omega) \, d^2\omega := \langle \omega | \rho | \omega \rangle \frac{d^2\omega}{\pi}, \qquad (1)$$

where a coherent state $|\omega\rangle$ is defined as

$$|\omega\rangle := e^{-|\omega|^2/2} \sum_{n=0}^{\infty} \frac{\omega^n}{\sqrt{n!}} |n\rangle. \qquad (2)$$

We refer to the expectation associated with the distribution $q_\rho(\omega)$ simply as $\mathbb{E}_\rho$. To construct a lower bound for the fidelity $\langle 0|\rho|0 \rangle$ from $\hat{\omega}$, we will use the associated Laguerre polynomials which are given by

$$L_n^{(k)}(\nu) := (-1)^k \frac{d^k L_{n+k}(\nu)}{d\nu^k}, \qquad (3)$$

where

$$L_n(\nu) := \frac{e^\nu}{n!} \frac{d^n}{d\nu^n} (e^{-\nu} \nu^n) \qquad (4)$$

are the Laguerre polynomials. Our test scheme is based on the following theorem.

*Theorem 1*: Let $\Lambda_{m,r}(\nu)$ $(\nu \geq 0)$ be a bounded function given by

$$\Lambda_{m,r}(\nu) := e^{-r\nu}(1+r)L_m^{(1)}((1+r)\nu), \qquad (5)$$

for an integer $m \geq 0$ and a real number $r > 0$. Then, we have

$$\mathbb{E}_\rho[\Lambda_{m,r}(|\hat{\omega}|^2)] = \langle 0|\rho|0 \rangle + \sum_{n=m+1}^{\infty} \frac{\langle n|\rho|n \rangle}{(1+r)^n} I_{n,m}, \qquad (6)$$

where $I_{n,m}$ are constants satisfying $(-1)^m I_{n,m} > 0$.

From Eq. (6), a lower bound on the fidelity between $\rho$ and the vacuum state is given by

$$\mathbb{E}_\rho[\Lambda_{m,r}(|\hat{\omega}|^2)] \leq \langle 0|\rho|0 \rangle \quad (m : \text{odd}) \qquad (7)$$

for any odd integer $m$. As seen in Fig. 1a, the absolute value and the slope of the function $\Lambda_{m,r}$ are moderate for small values of $m$ and $r$, which is advantageous in executing the test in a finite duration with a finite resolution. Compared to a similar method proposed in ref. [39], our method excels in its tightness for weak input signals; we see from Eq. (6) that, regardless of the value of $r$, the inequality (7) saturates when $\rho$ has at most $m$ photons. This is crucial for the use in QKD in which tightness directly affects the efficiency of the key generation.

Extension to the fidelity to a coherent state $|\beta\rangle$ is straightforward as

$$\mathbb{E}_\rho[\Lambda_{m,r}(|\hat{\omega} - \beta|^2)] \leq \text{Tr}(\rho|\beta\rangle \langle \beta|) \quad (m : \text{odd}). \qquad (8)$$

The proofs are given in Methods.

**Proposed protocol.** Based on this fidelity test, we propose the following discrete-modulation protocol (see Fig. 2). Prior to the protocol, Alice and Bob determine the number of rounds $N$, the acceptance probability of homodyne measurement $f_{\text{suc}}(|x|)$ $(x \in \mathbb{R})$ with $f_{\text{suc}}(0) = 0$, the parameters for the test function $(m, r)$, and the protocol parameters $(\mu, p_{\text{sig}}, p_{\text{test}}, p_{\text{trash}}, \beta, s)$ with $p_{\text{sig}} + p_{\text{test}} + p_{\text{trash}} = 1$, where all the parameters are positive. Alice

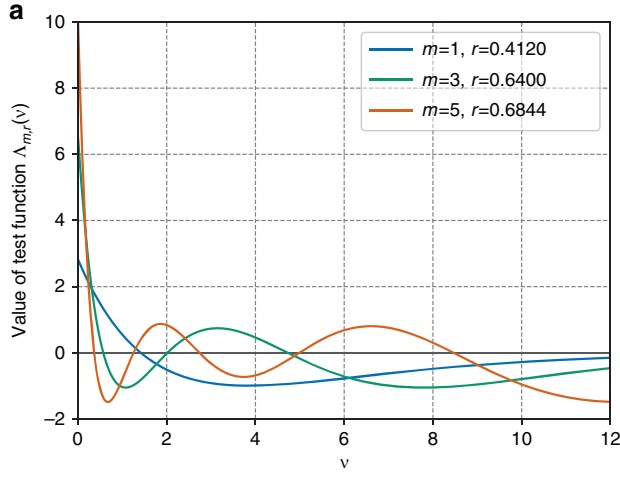

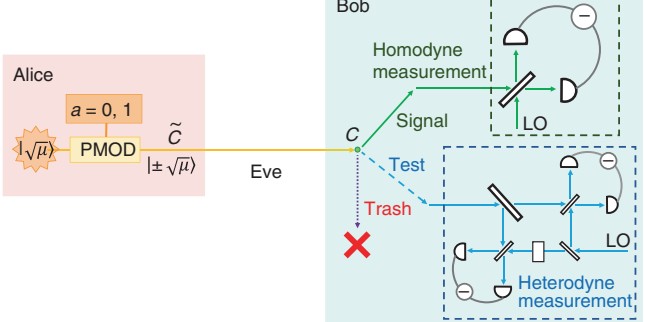

**Fig. 2 The proposed continuous-variable quantum key distribution protocol.** Alice generates a random bit $a \in \{0, 1\}$ and sends a coherent state with amplitude $(-1)^a \sqrt{\mu}$. Bob chooses one of the three measurements based on the predetermined probability. In the signal round, Bob performs a homodyne measurement on the received optical pulse and obtains an outcome $\hat{x}$. In the test round, Bob performs a heterodyne measurement on the received optical pulse and obtains an outcome $\hat{\omega}$. In the trash round, he produces no outcome.

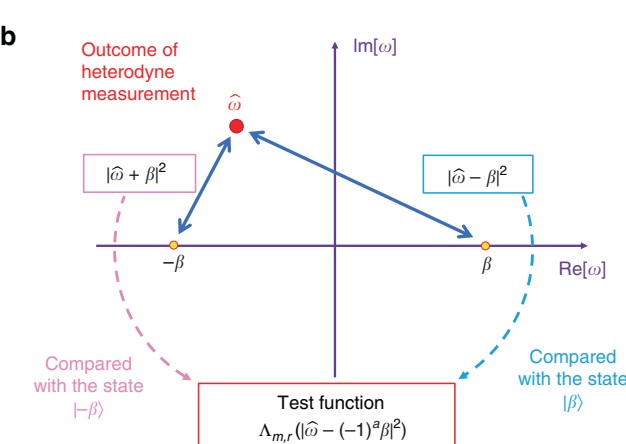

**Fig. 1 The test scheme to estimate the fidelity. a** Example of the test functions $\Lambda_{m,r}$ used in the estimation. The values of $r$ in the figure are chosen so that the range of $\Lambda_{m,r}$ is minimized for given $m$. In general, the minimum range of the function $\Lambda_{m,r}$ becomes larger as $m$ increases. The pair $(m, r) = (1, 0.4120)$ was used in the numerical simulation of key rates below. **b** A schematic description of the usage of obtained outcomes in heterodyne measurement. In order to estimate the lower bound on the fidelity to the coherent states $|\pm\beta\rangle$, the squared distance between the outcome $\hat{\omega}$ and the objective point $(-1)^a\beta$ (i.e., $|\hat{\omega} - (-1)^a\beta|^2$) is used.

and Bob then run the protocol described in Box 1. Upon successful verification, the protocol generates a shared final key of length

$$\hat{N}^{\mathrm{fin}} = \hat{N}^{\mathrm{suc}}\left(1 - h\left(U(\hat{F}, \hat{N}^{\mathrm{trash}})/\hat{N}^{\mathrm{suc}}\right)\right) - s \qquad (9)$$

where $h(x) := -x\log_2(x) - (1-x)\log_2(1-x)$ is the binary entropy function and the function $U(\hat{F}, \hat{N}^{\mathrm{trash}})$ will be specified later.

The acceptance probability $f_{\mathrm{suc}}(|x|)$ should be chosen to post-select the rounds with larger values of $|x|$, for which the bit error probability is expected to be lower. It is ideally a step function, but our security proof is applicable to any form of $f_{\mathrm{suc}}(|x|)$. The parameter $\beta$ is typically chosen to be $\sqrt{\eta\mu}$ with $\eta$ being a nominal transmissivity of the quantum channel, while the security proof itself holds for any choice of $\beta$. The parameters $s$ and $s'$ are related to the overall security parameter in the security proof below.

**Security proof.** We determine a sufficient amount of the privacy amplification according to Shor and Preskill[37,40], which has been widely used for the DV-QKD protocols. We consider a coherent version of Steps 1 and 2, in which Alice and Bob share an entangled pair of qubits for each success signal round, such that

their $Z$-basis measurement outcomes correspond to the sifted key bits $a$ and $b$. For Alice, we introduce a qubit $A$ and assume that she entangles it with an optical pulse $\tilde{C}$ in a state

$$|\Psi\rangle_{A\tilde{C}} := \frac{|0\rangle_A |\sqrt{\mu}\rangle_{\tilde{C}} + |1\rangle_A |-\sqrt{\mu}\rangle_{\tilde{C}}}{\sqrt{2}}. \qquad (10)$$

Then, Step 1 is equivalent to the preparation of $|\Psi\rangle_{A\tilde{C}}$ followed by a measurement of the qubit $A$ on $Z$ basis $\{|0\rangle, |1\rangle\}$ to determine the bit value $a$. For Bob, we construct a process of probabilistically converting the received optical pulse $C$ to a qubit $B$ (See Fig. 3). Consider a completely positive (CP) map defined by

$$\mathcal{F}_{C\to B}(\rho_C) := \int_0^\infty dx\, K^{(x)} \rho_C K^{(x)\dagger} \qquad (11)$$

with

$$K^{(x)} := \sqrt{f_{\mathrm{suc}}(x)}\left(|0\rangle_B \langle x|_C + |1\rangle_B \langle -x|_C\right), \qquad (12)$$

where $\langle x|$ maps a state vector to the value of its wave function at $x$ (See also Eq. (111)). When the pulse $C$ is in a state $\rho_C$, the corresponding process succeeds with a probability $p_{\mathrm{suc}}$ and then prepares the qubit $B$ in a state $\rho_B$, where $p_{\mathrm{suc}}\rho_B = \mathcal{F}_{C\to B}(\rho_C)$. If the qubit $B$ is further measured on $Z$ basis, probabilities of the outcome $b = 0, 1$ are given by

$$p_{\mathrm{suc}}\langle 0|\rho_B|0\rangle = \int_0^\infty f_{\mathrm{suc}}(x)dx\, \langle x|\rho_C|x\rangle, \qquad (13)$$

$$p_{\mathrm{suc}}\langle 1|\rho_B|1\rangle = \int_0^\infty f_{\mathrm{suc}}(x)dx\, \langle -x|\rho_C|-x\rangle, \qquad (14)$$

which shows the equivalence to the signal round in Step 2. This is illustrated in Fig. 3.

To clarify the above observation, we introduce an entanglement-sharing protocol defined in Box 2. This protocol leaves $\hat{N}^{\mathrm{suc}}$ pairs of qubits shared by Alice and Bob. If they measure these qubits on $Z$ basis to define the sifted key bits, the whole procedure is equivalent to Steps 1 through 3 of the actual protocol (see Fig. 4). Alice's measurements on $X$ basis $\{|\pm\rangle := (|0\rangle + |1\rangle)/\sqrt{2}\}$ in the trash rounds are added for later security argument, and they do not affect the equivalence.

The Shor-Preskill argument connects the amount of privacy amplification to the so-called phase error rate. Suppose that, after

---

**Box 1 | Actual protocol**

1. Alice generates a random bit $a \in \{0, 1\}$ and sends an optical pulse $\tilde{C}$ in a coherent state with amplitude $(-1)^a \sqrt{\mu}$ to Bob. She repeats it $N$ times.

2. For each of the received $N$ pulses, Bob chooses a label from {signal, test, trash} with probabilities $p_{sig}$, $p_{test}$, and $p_{trash}$, respectively. According to the label, Alice and Bob do one of the following procedures.

[signal] Bob performs a homodyne measurement on the received optical pulse $C$, and obtains an outcome $\hat{x} \in \mathbb{R}$. With a probability $f_{suc}(|\hat{x}|)$, he regards the detection to be a "success", and defines a bit $b = 0$ (resp. 1) when $\text{sign}(\hat{x}) = +(-)1$. He announces success/failure of the detection. In the case of a success, Alice (resp. Bob) keeps $a$ ($b$) as a sifted key bit.

[test] Bob performs a heterodyne measurement on the received optical pulse $C$, and obtains an outcome $\hat{\omega}$. Alice announces her bit $a$. Bob calculates the value of $\Lambda_{m,r}(|\hat{\omega} - (-1)^a \beta|^2)$.

[trash] Alice and Bob produce no outcomes.

3. We refer to the numbers of "success" and "failure" signal rounds, test rounds, and trash rounds as $\hat{N}^{suc}$, $\hat{N}^{fail}$, $\hat{N}^{test}$, and $\hat{N}^{trash}$, respectively. ($N = \hat{N}^{suc} + \hat{N}^{fail} + \hat{N}^{test} + \hat{N}^{trash}$ holds by definition.) Bob calculates the sum of $\Lambda_{m,r}(|\hat{\omega} - (-1)^a \beta|^2)$ obtained in the $\hat{N}^{test}$ test rounds, which is denoted by $\hat{F}$.

4. For error correction, they use $(H_{EC} + s')$-bits of encrypted communication consuming a pre-shared secret key to do the following. Alice sends Bob $H_{EC}$ bits of syndrome of a linear code for her sifted key. Bob reconciles his sifted key accordingly. Alice and Bob verify the correction by comparing $s'$ bits via universal$_2$ hashing[54].

5. Bob computes and announces the final key length $\hat{N}^{fin}$ according to Eq. (9). Alice and Bob apply privacy amplification to obtain the final key. The net key gain $\hat{G}$ per pulse is thus given by $\hat{G} = (\hat{N}^{fin} - H_{EC} - s')/N$.

---

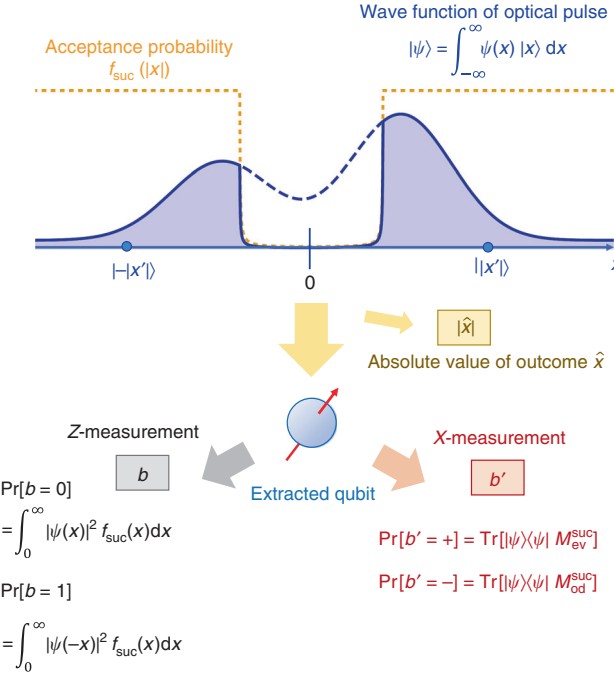

**Fig. 3 Bob's qubit extraction in the entanglement-sharing protocol.** Bob performs on the optical pulse a non-demolition projective measurement, with which the absolute value of the outcome of homodyne measurement $|\hat{x}|$ is determined. Then, Bob extracts a qubit $B$ by the operation $\mathcal{F}$ defined in Eq. (11). A $Z$-basis measurement on this qubit gives the same sifted key bit $b$ as described in the actual protocol. On the other hand, the $X$-basis measurement on this qubit reveals the parity of photon number of the received optical pulse.

the entanglement-sharing protocol, Alice and Bob measure their $\hat{N}^{suc}$ pairs of qubits on $X$ basis $\{|+\rangle, |-\rangle\}$. A pair with outcomes $(+, -)$ or $(-, +)$ is defined to be a phase error. Let $\hat{N}^{suc}_{ph}$ be the number of phase errors among $\hat{N}^{suc}$ pairs. If we can have a good upper bound $e_{ph}$ on the phase error rate $\hat{N}^{suc}_{ph}/\hat{N}^{suc}$, shortening by fraction $h(e_{ph})$ via privacy amplification in the actual protocol achieves the security in the asymptotic limit[37].

To cover the finite-size case as well, we need a more rigorous statement on the upper bound. For that purpose, we define an

estimation protocol in Box 3 (see also Fig. 4). The task of proving the security of the actual protocol is then reduced to construction of a function $U(\hat{F}, \hat{N}^{trash})$ which satisfies

$$\Pr\left[\hat{N}^{suc}_{ph} \le U(\hat{F}, \hat{N}^{trash})\right] \ge 1 - \epsilon \qquad (15)$$

for any attack in the estimation protocol. It is known that the condition (15) immediately implies that the actual protocol is $\epsilon_{sec}$-secure with a small security parameter $\epsilon_{sec} = \sqrt{2}\sqrt{\epsilon + 2^{-s}} + 2^{-s'}$ [40,41]. See Methods for the rationale and the detailed definition of security.

At this point, it is beneficial for the analysis of the phase error statistics to clarify what property of the optical pulse $C$ is measured by Bob's $X$-basis measurement in the estimation protocol (see Fig. 3). Let $\Pi_{ev(od)}$ be the projection to the subspace with even (resp. odd) photon numbers. ($\Pi_{ev} + \Pi_{od} = \mathbf{1}_C$ holds by definition.) Furthermore, since $\Pi_{ev} - \Pi_{odd}$ is the operator for an optical phase shift of $\pi$, we have $(\Pi_{ev} - \Pi_{odd})|x\rangle = |-x\rangle$. Eq. (12) is then rewritten as

$$K^{(x)} = \sqrt{2f_{suc}(x)}\left(|+\rangle_B \langle x|_C \Pi_{ev} + |-\rangle_B \langle x|_C \Pi_{od}\right). \qquad (16)$$

Therefore, when the state of the pulse $C$ is $\rho_C$, the probability of obtaining $+(-)$ in the $X$-basis measurement in the estimation protocol is given by

$$\langle +(-)|\mathcal{F}_{C\to B}(\rho_C)|+(-)\rangle = \text{Tr}\left(\rho_C M^{suc}_{ev(od)}\right), \qquad (17)$$

where

$$M^{suc}_{ev(od)} := \int_0^\infty 2f_{suc}(x)dx \, \Pi_{ev(od)}|x\rangle \langle x|_C \Pi_{ev(od)}. \qquad (18)$$

This shows that Bob's $X$-basis measurement distinguishes the parity of the photon number of the received pulse. In this sense, the secrecy of our protocol is assured by the complementarity between the sign of the quadrature and the parity of the photon number.

As an intermediate step toward our final goal of Eq. (15), let us first derive a bound on the expectation value $\mathbb{E}[\hat{N}^{suc}_{ph}]$ in terms of those collected in the test and the trash rounds, $\mathbb{E}[\hat{F}]$ and $\mathbb{E}[\hat{Q}_-]$, in the estimation protocol. Let $\rho_{AC}$ be the state of the qubit $A$ and the received pulse $C$ averaged over $N$ pairs, and define relevant operators as

$$M^{suc}_{ph} := |+\rangle \langle +|_A \otimes M^{suc}_{od} + |-\rangle \langle -|_A \otimes M^{suc}_{ev}, \qquad (19)$$

---

**Box 2 ❘ Entanglement-sharing protocol**

1′. Alice prepares a qubit $A$ and an optical pulse $\tilde{C}$ in a state $|\Psi\rangle_{A\tilde{C}}$ defined in (10). She repeats it $N$ times.

2′. For each of the received $N$ pulses, Bob announces a label in the same way as that in Step 2. Alice and Bob do one of the following procedures according to the label.

[signal] Bob performs a quantum operation on the received pulse $C$ specified by the CP map $\mathcal{F}_{C\to B}$ to determine success/failure of detection and to obtain a qubit $B$ upon success. He announces success/failure of detection. In the case of a success, Alice keeps her qubit $A$.

[test] Bob performs a heterodyne measurement on the received optical pulse $C$, and obtains an outcome $\hat{\omega}$. Alice measures her qubit $A$ on $Z$ basis and announces the outcome $a \in \{0, 1\}$. Bob calculates the value of $\Lambda_{m,r}(|\hat{\omega} - (-1)^a\beta|^2)$.

[trash] Alice measures her qubit $A$ on $X$ basis to obtain $a' \in \{+, -\}$.

3′. $\hat{N}^{\mathrm{suc}}, \hat{N}^{\mathrm{fail}}, \hat{N}^{\mathrm{test}}, \hat{N}^{\mathrm{trash}}$, and $\hat{F}$ are defined in the same way as those in Step 3. Let $\hat{Q}_-$ be the number of rounds in the $\hat{N}^{\mathrm{trash}}$ trash rounds with $a' = -$.

---

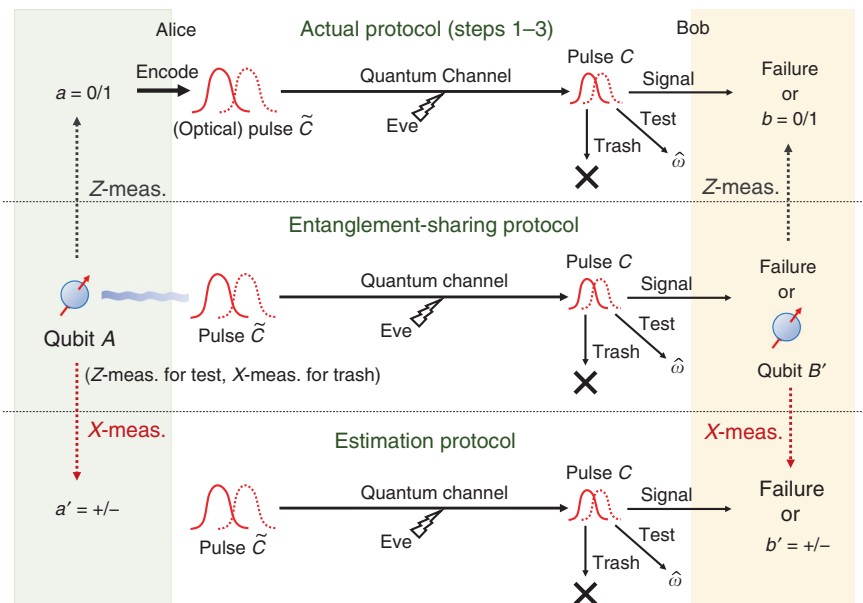

**Fig. 4 Relation between three protocols.** The actual protocol and the estimation protocol are related through the entanglement-sharing protocol. After the entanglement-sharing protocol, Alice and Bob are left with the observed data $(\hat{N}^{\mathrm{suc}}, \hat{N}^{\mathrm{fail}}, \hat{N}^{\mathrm{test}}, \hat{N}^{\mathrm{trash}}, \hat{F}, \hat{Q}_-)$ and $\hat{N}^{\mathrm{suc}}$ pairs of qubits. If Alice and Bob ignore $\hat{Q}_-$ and measure their qubits on the $Z$-basis to determine their $\hat{N}^{\mathrm{suc}}$-bit sifted keys, it becomes equivalent to the actual protocol. On the other hand, if Alice and Bob measure their $\hat{N}^{\mathrm{suc}}$ pairs of qubits on the $X$-basis, they can count the number $\hat{N}^{\mathrm{suc}}_{\mathrm{ph}}$ of phase errors, which we call the estimation protocol. If we can find a reliable upper bound $U$ on $\hat{N}^{\mathrm{suc}}_{\mathrm{ph}}$ in the estimation protocol, it restricts the property of the state of $\hat{N}^{\mathrm{suc}}$ pairs of qubits after the entanglement-sharing protocol, which in turn limits the amount of leaked information on the sifted keys in the actual protocol. The security proof is thus reduced to finding such an upper bound $U$ in the estimation protocol, represented as a function of the variables that are commonly available in the three protocols.

---

**Box 3 ❘ Estimation protocol**

1″–3″. Same as Steps 1′, 2′, and 3′ of the entanglement-sharing protocol.

4″. Alice and Bob measure each of their $\hat{N}^{\mathrm{suc}}$ pairs of qubits on $X$ basis and obtain outcomes $a'$ and $b'$, respectively. Let $\hat{N}^{\mathrm{suc}}_{\mathrm{ph}}$ be the number of pairs found in the combination $(a', b') = (+, -)$ or $(-, +)$.

---

$$\Pi^{\mathrm{fid}} := |0\rangle\langle 0|_A \otimes |\beta\rangle\langle\beta|_C + |1\rangle\langle 1|_A \otimes |-\beta\rangle\langle-\beta|_C, \quad (20)$$

$$\Pi^{\mathrm{trash}}_- := |-\rangle\langle-|_A \otimes \mathbf{1}_C. \quad (21)$$

Then we immediately have

$$\mathbb{E}[\hat{N}^{\mathrm{suc}}_{\mathrm{ph}}] = p_{\mathrm{sig}} N \operatorname{Tr}\left(\rho_{AC} M^{\mathrm{suc}}_{\mathrm{ph}}\right) \quad (22)$$

and

$$\mathbb{E}[\hat{Q}_-] = p_{\mathrm{trash}} N \operatorname{Tr}\left(\rho_{AC}\Pi^{\mathrm{trash}}_-\right), \quad (23)$$

while application of the property of Eq. (8) leads to

$$\mathbb{E}[\hat{F}] \le p_{\mathrm{test}} N \operatorname{Tr}\left(\rho_{AC}\Pi^{\mathrm{fid}}\right). \quad (24)$$

Let us denote $\operatorname{Tr}(\rho_{AC}M)$ simply by $\langle M\rangle$ for any operator $M$. The set of points $\left(\langle M^{\mathrm{suc}}_{\mathrm{ph}}\rangle, \langle\Pi^{\mathrm{fid}}\rangle, \langle\Pi^{\mathrm{trash}}_-\rangle\right)$ for all the density operators $\rho_{AC}$ form a convex region. Rather than directly deriving the boundary of the region, it is easier to pursue linear constraints in the form of

$$\left\langle M^{\mathrm{suc}}_{\mathrm{ph}}\right\rangle \le B(\kappa, \gamma) - \kappa\left\langle\Pi^{\mathrm{fid}}\right\rangle + \gamma\left\langle\Pi^{\mathrm{trash}}_-\right\rangle, \quad (25)$$

where $B(\kappa, \gamma), \kappa, \gamma \in \mathbb{R}$.

It is expected that a meaningful bound is obtained only for $\kappa, \gamma \geq 0$. Decreasing fidelity $\langle \Pi^{\text{fid}} \rangle$ should allow more room for eavesdropping, leading to a larger value of phase error rate $\langle M_{\text{ph}}^{\text{suc}} \rangle$. Hence Eq. (25) will give a good bound only when $\kappa \geq 0$. As for $\langle \Pi_{-}^{\text{trash}} \rangle$, it only depends on the marginal state of Alice's qubit $A$, which is independent of the adversary's attack. We thus have $\langle \Pi_{-}^{\text{trash}} \rangle = q_{-} := \| \langle -|_A |\Psi\rangle_{A\hat{C}} \|^2 = (1 - e^{-2\mu})/2$. Since Alice's use of a stronger pulse should lead to larger leak of information, we should choose $\gamma \geq 0$ for a good bound.

To find a function $B(\kappa, \gamma)$ satisfying Eq. (25), let us define an operator

$$M[\kappa, \gamma] := M_{\text{ph}}^{\text{suc}} + \kappa \Pi^{\text{fid}} - \gamma \Pi_{-}^{\text{trash}}. \qquad (26)$$

Then Eq. (25) is rewritten as $\text{Tr}\big(\rho_{AC} M[\kappa, \gamma]\big) \leq B(\kappa, \gamma)$. This condition holds for all $\rho_{AC}$ iff $M[\kappa, \gamma]$ satisfies an operator inequality

$$M[\kappa, \gamma] \leq B(\kappa, \gamma) \mathbf{1}_{AC}. \qquad (27)$$

If the operator $M[\kappa, \gamma]$ was represented by a matrix of a small size, the tightest bound would be found by computing the maximum eigenvalue of the matrix. But here $M[\kappa, \gamma]$ has an infinite rank and it is difficult to compute the tightest bound. We thus compromise and heuristically find a computable bound $B(\kappa, \gamma)$ which is not necessarily tight; we reduce the problem to finding the maximum eigenvalues of small-size matrices by replacing $M[\kappa, \gamma]$ with a constant upper-bound except in a relevant finite-dimensional subspace spanned by $|\pm\beta\rangle$ and $M_{\text{ev(od)}}^{\text{suc}} |\pm\beta\rangle$. For the detailed derivation of $B(\kappa, \gamma)$, see Methods.

With $B(\kappa, \gamma)$ computed, we can rewrite Eq. (25) using Eqs. (22)–(24) to obtain a relation between $\mathbb{E}[\hat{N}_{\text{ph}}^{\text{suc}}]$, $\mathbb{E}[\hat{F}]$, and $\mathbb{E}[\hat{Q}_{-}]$. It is concisely written as

$$\mathbb{E}\big[\hat{T}[\kappa, \gamma]\big] \leq NB(\kappa, \gamma) \qquad (28)$$

with

$$\hat{T}[\kappa, \gamma] := p_{\text{sig}}^{-1} \hat{N}_{\text{ph}}^{\text{suc}} + p_{\text{test}}^{-1} \kappa \hat{F} - p_{\text{trash}}^{-1} \gamma \hat{Q}_{-}. \qquad (29)$$

This relation leads to an explicit bound on the phase error rate as $\mathbb{E}[\hat{N}_{\text{ph}}^{\text{suc}}]/p_{\text{sig}} N \leq B(\kappa, \gamma) + \gamma q_{-} - \kappa \, \mathbb{E}[\hat{F}]/p_{\text{test}} N$, which is enough for the computation of asymptotic key rates.

The security in the finite-size regime is proved as follows. The fact that the bound given in Eq. (28) is true for all the states $\rho_{AC}$ allows us to use Azuma's inequality[42] to evaluate the fluctuations around the expectation value, leading to an inequality

$$\hat{T}[\kappa, \gamma] \leq NB(\kappa, \gamma) + \delta_1(\epsilon/2) \qquad (30)$$

which holds with a probability no smaller than $1 - \epsilon/2$ (see Methods for the explicit form of $\delta_1(\epsilon/2)$, which is of $O(\sqrt{N})$). We remark that the reason for including the trash rounds in the actual protocol is to circumvent a technical issue which would arise in this step. Without measurement of $\hat{Q}_{-}$ in the estimation protocol, we would obtain an inequality $\mathbb{E}[p_{\text{sig}}^{-1} \hat{N}_{\text{ph}}^{\text{suc}} + p_{\text{test}}^{-1} \kappa \hat{F}] \leq NB(\kappa, \gamma) + \gamma q_{-}$. In contrast to Eq. (28), the new inequality is true only when $\rho_{AC}$ satisfies $\langle \Pi_{-}^{\text{trash}} \rangle = q_{-}$, which is too stringent for the application of Azuma's inequality.

Although Eq. (29) includes $\hat{Q}_{-}$ which is inaccessible in the actual protocol, we can derive a bound by noticing that it is an outcome from Alice's qubits and is independent of the adversary's attack. In fact, given $\hat{N}^{\text{trash}}$, it is the tally of $\hat{N}^{\text{trash}}$ Bernoulli trials with a probability $q_{-}$. Hence, we can derive an inequality of the

form

$$\hat{Q}_{-} \leq q_{-} \hat{N}^{\text{trash}} + \delta_2(\epsilon/2; \hat{N}^{\text{trash}}) \qquad (31)$$

which holds with a probability no smaller than $1 - \epsilon/2$. Here $\delta_2(\epsilon/2; \hat{N}^{\text{trash}})$ can be determined by a Chernoff bound (see Methods). Combining Eqs. (29), (30), and (31), we obtain $U(\hat{F}, \hat{N}^{\text{trash}})$ satisfying Eq. (15) to complete the finite-size security proof.

**Numerical simulation.** We simulated the net key gain per pulse $\hat{G}$ as a function of attenuation in the optical channel (including the efficiency of Bob's apparatus). We assume a channel model with a loss with transmissivity $\eta$ and an excess noise at channel output; Bob receives Gaussian states obtained by randomly displacing coherent states $|\pm\sqrt{\eta\mu}\rangle$ to increase their variances by a factor of $(1 + \xi)$[43,44]. We assume a step function with a threshold $x_{\text{th}}( > 0)$ as the acceptance probability $f_{\text{suc}}(|x|)$. The expected amplitude of coherent state $\beta$ is chosen to be $\sqrt{\eta\mu}$. We set $\epsilon_{\text{sec}} = 2^{-50}$ for the security parameter, and set $\epsilon = 2^{-s} = \epsilon_{\text{sec}}^2/16$ and $2^{-s'} = \epsilon_{\text{sec}}/2$. We thus have two coefficients $(\kappa, \gamma)$, four protocol parameters $(\mu, x_{\text{th}}, p_{\text{sig}}, p_{\text{test}})$, and two parameters $(m, r)$ of the test function to be determined. For each transmissivity $\eta$, we determined $(\kappa, \gamma)$ via a convex optimization using the CVXPY 1.0.25[45,46] and $(\mu, x_{\text{th}}, p_{\text{sig}}, p_{\text{test}})$ via the Nelder-Mead in the scipy.minimize library in Python, in order to maximize the key rate. Furthermore, we adopted $m = 1$ and $r = 0.4120$, which leads to $(\max \Lambda_{m,r}, \min \Lambda_{m,r}) = (2.824, -0.9932)$. See Methods for the detail of the model of our numerical simulation and examples of optimized parameters. Typical optimized values of the threshold $x_{\text{th}}$ range from 0.4 to 1.5 (we adopted a normalization for which the vacuum fluctuation is $\sqrt{\langle (\Delta x)^2 \rangle} = 0.5$). They are larger than those in other analyses of protocols with post-selection (e.g., ref. [31]). A possible reason is the fact that the latter protocols use more than two states to monitor the eavesdropping act, which may lead to a lower cost of privacy amplification and higher tolerance against bit errors.

Figure 5 shows the key rates of our protocol in the asymptotic limit $N \to \infty$ and finite-size cases with $N = 10^9$–$10^{12}$ for $\xi = 10^{-2.0}$–$10^{-3.0}$ and 0. (Note that from the results of the recent experiments[8,44,47], excess noise with $\xi = 10^{-2.0}$–$10^{-3.0}$ at the channel output seems reasonable. Furthermore, the state-of-the-art experiments[8] work at 0.5 GHz repetition rate, which implies that total number of rounds $N = 10^9$–$10^{12}$ can be achieved in a realistic duration.) For the noiseless model ($\xi = 0$), the asymptotic rate reaches 8 dB. In the case of $\xi = 10^{-3.0}$, it reaches 4 dB, which is comparable to the result of a similar binary modulation protocol proposed in ref. [29]. As for finite-size key rates, we see that the noiseless model shows a significant finite-size effect even for $N = 10^{12}$. On the other hand, with a presence of noises ($\xi = 10^{-3.0}$) the effect becomes milder, and $N = 10^{11}$ is enough to achieve a rate close to the asymptotic case. This may be ascribed to the cost of the fidelity test. In order to make sure that the fidelity is no smaller than $1 - \delta$, the statistical uncertainty of the fidelity test must be reduced to $O(\delta)$. As a result, approaching the asymptotic rate of $\xi = 0$ will require many rounds for the fidelity test.

**Discussion**

Numerically simulated key rates above were computed on the implicit assumption that Bob's observed quantities are processed with infinite precision. Even when these are approximated with a finite set of discrete points, we can still prove the security with minimal degradation of key rates. For the heterodyne

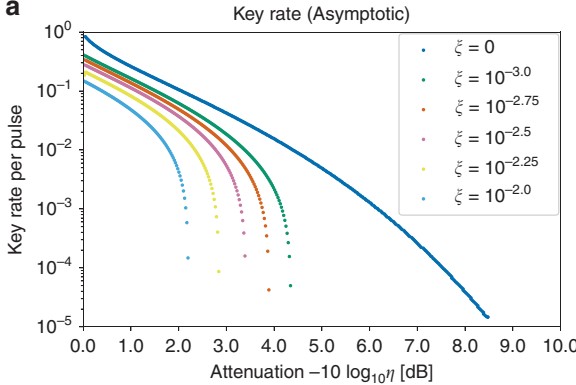

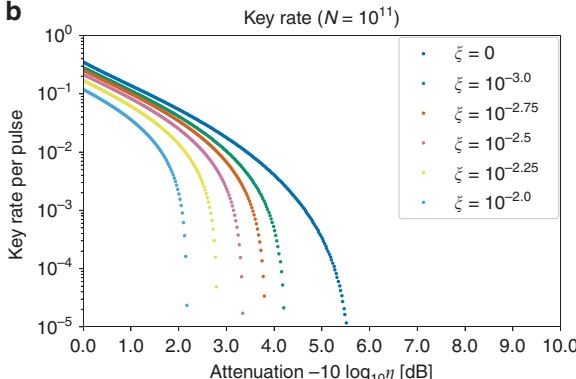

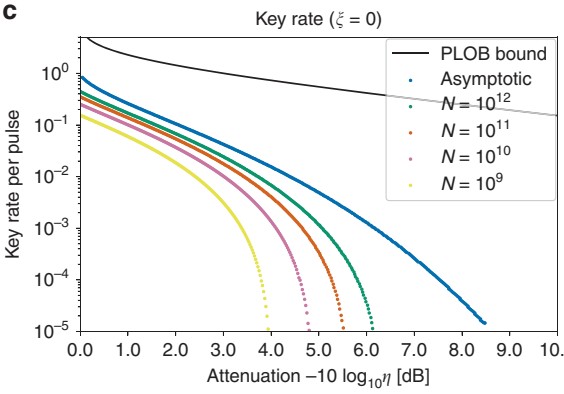

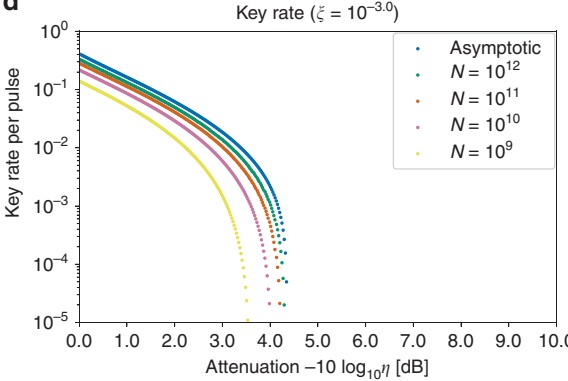

**Fig. 5 The net key gain per pulse $\hat{G}$ (key rate) vs. transmissivity $\eta$ of the optical channel.** The abscissa represents attenuation in decibel, i.e., $-10 \log_{10} \eta$. We assumed that the optical pulse that Bob receives is given by randomly displacing a coherent state to increase its variance by a factor of $(1 + \xi)$. **a** The asymptotic key rate for various values of $\xi$. **b** The key rate for various values of $\xi$ when the pulse number is finite ($N = 10^{11}$). **c** The key rate without the excess noise ($\xi = 0$) along with the repeaterless bound (PLOB bound) of the secure key rate in the pure-loss channel[55]. **d** The key rate with the excess noise of $\xi = 10^{-3.0}$.

in Fig. 1a, the slope of function $\Lambda_{m,r}(v)$ is moderate and goes to zero for $v \to \infty$. This means that the worst-case value can be made close to the true value, leading to small influence on the key rate. For the homodyne measurement used for the signal, finite precision can be treated through appropriate modification of the acceptance probability $f_{\text{suc}}(x)$. Aside from a very small change in the success rate and the bit error rate, this function affects the key rate only through integrals in Eqs. (116), (118), and (120) in Methods, and hence influence on the key rate is expected to be small. We thus believe that the fundamental obstacles associated with the analog nature of the CV protocol have been settled by our approach.

In comparison with recent asymptotic analyses[31,32] of discrete-modulation CV QKD, our protocol achieves lower key rates and much shorter distance. Since ours is the first attempt of applying the proof technique of DV QKD to CV QKD, there is much room for possible improvement. We sacrificed the optimality for simplicity in deriving the operator inequality. The definition of the phase error is not unique and there may be a better choice. The trash rounds were introduced for technical reasons, but we are not sure whether they are really necessary. Nonetheless, we believe that the dominant reason for the difference lies in the fact that our protocol uses only two states. In contrast, the protocols considered in refs. [31,32] use four or more states in signal or test modes. The genuine binary protocol was analyzed in ref. [29], and the key rate derived there is comparable to ours.

In order to improve the presented finite-size key rate, a promising route will thus be increasing the number of states from two. Our fidelity test can be straightforwardly generalized to monitoring of such a larger constellation of signals, and we will be able to confine the adversary's attacks more tightly than in the present binary protocol. As for the proof techniques to determine the amount of privacy amplification, there are two possible directions. One is to generalize the present DV-QKD-inspired approach of estimating the number of phase errors in qubits to the case of qudits. The other direction is to seek a way to combine the existing analyses[31,32,48] of discrete-modulation CV-QKD protocols, which have been reported to yield high key rates in the asymptotic regime, to our fidelity test. Although either of the approaches is nontrivial, we believe that the present results will open up new direction toward exploiting the expected high potential of CV QKD with an improved security level.

In summary, we proved the security of a binary-modulated CV-QKD protocol in the finite-size regime while completely circumventing the problems arising from the analog nature of CV QKD. We believe that it is a significant milestone toward real-world implementation of CV QKD, which has its own advantages.

## Methods

**Proof of Theorem 1 and Eq. (8).** In this section, we prove Theorem 1 stated in the main text and derive Eq. (8) as a corollary of Theorem 1.

measurement used for the test in the protocol, assume that a digitized outcome $\omega_{\text{dig}}$ ensures that the true value $\hat{\omega}$ lies in a range $\Omega(\omega_{\text{dig}})$. Then, we need only to replace $\Lambda_{m,r}(|\hat{\omega} \pm \beta|^2)$ with its worst-case value, $\min\{\Lambda_{m,r}(|\hat{\omega} \pm \beta|^2) : \hat{\omega} \in \Omega(\omega_{\text{dig}})\}$. As seen

*Proof*: From Eq. (1), the expectation value of $\Lambda_{m,r}(|\hat{\omega}|^2)$ when given a measured state $\rho$ is given by

$$
\begin{aligned}
&\mathbb{E}_\rho[\Lambda_{m,r}(|\hat{\omega}|^2)] \\
&= \int_{\omega \in \mathbb{C}} \Lambda_{m,r}(|\omega|^2) q_\rho(\omega) \, d^2\omega \\
&= \int_0^\infty d\nu \, \Lambda_{m,r}(\nu) \left( \int_0^{2\pi} \frac{d\theta}{2\pi} \left\langle \sqrt{\nu}e^{i\theta} \middle| \rho \middle| \sqrt{\nu}e^{i\theta} \right\rangle \right) \\
&= \int_0^\infty d\nu \, \Lambda_{m,r}(\nu) \left( \sum_{n=0}^\infty \frac{\nu^n e^{-\nu}}{n!} \langle n|\rho|n\rangle \right) \\
&= \sum_{n=0}^\infty \frac{\langle n|\rho|n\rangle I_{n,m}}{(1+r)^n},
\end{aligned}
\tag{32}
$$

where

$$
I_{n,m} := \frac{1}{n!} \int_0^\infty d\nu \, e^{-\nu} \nu^n L_m^{(1)}(\nu)
\tag{33}
$$

for integers $n, m \geq 0$.

The following three properties hold for $I_{n,m}$:

(i) $I_{n,m} = 0$ for $m \geq n \geq 1$.

This results from orthogonality relations of the associated Laguerre polynomials, that is,

$$
\int_0^\infty L_n^{(1)}(\nu) L_m^{(1)}(\nu) \nu e^{-\nu} \, d\nu = (n+1)\delta_{n,m}.
\tag{34}
$$

Since the polynomial $\nu^{n-1}$ can be written as a linear combination of lower order polynomials $\{L_l^{(1)}(\nu)\}_{0 \leq l \leq n-1}$, $I_{n,m}$ vanishes whenever $m \geq n \geq 1$.

(ii) $(-1)^m I_{n,m} > 0$ for $n > m \geq 0$.

This property is shown as follows. First, the associated Laguerre polynomials satisfy the following recurrence relation for $m \geq 1$[49]:

$$
m L_m^{(1)}(\nu) = \nu \frac{dL_m^{(1)}}{d\nu}(\nu) + (m+1) L_{m-1}^{(1)}(\nu).
\tag{35}
$$

Substituting this to Eq. (33) and using integration by parts, we have

$$
I_{n,m} = \frac{n+m}{n} I_{n-1,m} - \frac{m+1}{n} I_{n-1,m-1}.
\tag{36}
$$

for $n \geq 1$ and $m \geq 1$. The property (ii) is then proved by induction over $m$. For $m = 0$, it is true since $I_{n,0} = 1 > 0$. When $(-1)^{m-1} I_{n,m-1} > 0$ for $n > m-1$, we can prove $(-1)^m I_{n,m} > 0$ for $n > m$ by using Eq. (36) recursively with $I_{m,m} = 0$ from property (i).

(iii) $I_{0,m} = 1$ for $m \geq 0$.

This also follows from property (i) and Eq. (36) for $n = 1$ and $m \geq 1$, which leads to $I_{0,m} = I_{0,0} = 1$.

Combining properties (i), (ii), and (iii) shows Eq. (6).

Eq. (8) in the main text is derived as the following corollary.

*Corollary 1*: Let $|\beta\rangle$ ($\beta \in \mathbb{C}$) be the coherent state with amplitude $\beta$. Then, for any $\beta \in \mathbb{C}$ and for any odd positive integer $m$, we have

$$
\mathbb{E}_\rho[\Lambda_{m,r}(|\hat{\omega} - \beta|^2)] \leq \langle \beta|\rho|\beta\rangle.
\tag{37}
$$

*Proof*: From Eq. (6) of Theorem 1, for any odd positive integer $m$, we have

$$
\mathbb{E}_\rho[\Lambda_{m,r}(|\hat{\omega}|^2)] \leq \langle 0|\rho|0\rangle.
\tag{38}
$$

Let $D_\beta$ be a displacement operator satisfying

$$
D_\beta|0\rangle\langle 0|D_\beta^\dagger = |\beta\rangle\langle\beta|,
\tag{39}
$$

and $D_\beta^\dagger = D_{-\beta}$. With $\tilde{\rho} := D_\beta \rho D_\beta^\dagger$, we have $q_{\tilde{\rho}}(\omega) = q_\rho(\omega - \beta)$ for probability density function of heterodyne measurement outcome, which implies that

$$
\begin{aligned}
\mathbb{E}_{\tilde{\rho}}[\Lambda_{m,r}(|\hat{\omega} - \beta|^2)] &= \mathbb{E}_\rho[\Lambda_{m,r}(|\hat{\omega}|^2)] \\
&\leq \langle 0|\rho|0\rangle \\
&= \langle \beta|\tilde{\rho}|\beta\rangle.
\end{aligned}
\tag{40}
$$

Replacing $\tilde{\rho}$ with $\rho$, we obtain Eq. (37).

**Detail of the security proof**. In this section, we prove the security of the proposed protocol in the main text. This section consists of several subsections. In the first subsection, we give a definition of security, which is standard in the literature. The security condition is divided into two conditions, secrecy and correctness. Since the correctness is trivially satisfied, it is the secrecy that is the focus of the security proof. The second subsection explains how the secrecy condition is reduced to Eq. (15) of

the estimation protocol, which bounds the number of phase errors. The third subsection lays the groundwork for the full security proof by deriving the inequality (27) involving three operators relevant for the quantities observed in the signal, the test, and the trash round in the estimation protocol. After proving a general lemma (Lemma 1), an explicit form of the upper bound $B(\kappa, \gamma)$ satisfying Eq. (27) is given as a corollary (Corollary 2) of the lemma. Finally, the fourth subsection uses Azuma's inequality and Corollaries 1 and 2 to derive an explicit form of $U(\hat{F}, \hat{N}^{\text{trash}})$ that fulfills Eq. (15), which completes the security proof of the actual protocol.

*1. Definition of security in the finite-size regime.* We evaluate the secrecy of the final key as follows. When the final key length is $N^{\text{fin}} \geq 1$, we represent Alice's final key and an adversary's quantum system as a joint state

$$
\rho_{\text{AE}|N^{\text{fin}}}^{\text{fin}} = \sum_{z=0}^{2^{N^{\text{fin}}}-1} \Pr(z)|z\rangle\langle z|_A \otimes \rho_{\text{E}|N^{\text{fin}}}^{\text{fin}}(z),
\tag{41}
$$

and define the corresponding ideal state as

$$
\rho_{\text{AE}|N^{\text{fin}}}^{\text{ideal}} = \sum_{z=0}^{2^{N^{\text{fin}}}-1} 2^{-N^{\text{fin}}} |z\rangle\langle z|_A \otimes \text{Tr}_A(\rho_{\text{AE}|N^{\text{fin}}}^{\text{fin}}).
\tag{42}
$$

Let $\|\sigma\|_1 = \text{Tr}\sqrt{\sigma^\dagger\sigma}$ be the trace norm of an operator $\sigma$. We say a protocol is $\epsilon_{\text{sct}}$-secret when

$$
\frac{1}{2} \sum_{N^{\text{fin}} \geq 1} \Pr(N^{\text{fin}}) \left\| \rho_{\text{AE}|N^{\text{fin}}}^{\text{fin}} - \rho_{\text{AE}|N^{\text{fin}}}^{\text{ideal}} \right\|_1 \leq \epsilon_{\text{sct}}
\tag{43}
$$

holds regardless of the adversary's attack. The main goal of the security proof is to derive the amount of privacy amplification, or equivalently to find the function $U(\hat{F}, \hat{N}^{\text{trash}})$ in Eq. (9), such that Eq. (43) should hold for given $\epsilon_{\text{sct}} > 0$.

For correctness, we say a protocol is $\epsilon_{\text{cor}}$-correct if the probability for Alice's and Bob's final key to differ is bounded by $\epsilon_{\text{cor}}$. Our protocol achieves $\epsilon_{\text{cor}} = 2^{-s'}$ via the verification in Step 4.

When the above two conditions are met, the protocol becomes $\epsilon_{\text{sec}}$-secure with $\epsilon_{\text{sec}} = \epsilon_{\text{sct}} + \epsilon_{\text{cor}}$ in the sense of universal composability[50].

*2. Reduction to the estimation protocol.* Here we show that Eq. (15) in the estimation protocol implies $\epsilon_{\text{sct}}$-secrecy of the actual protocol with $\epsilon_{\text{sct}} = \sqrt{2}\sqrt{\epsilon + 2^{-s}}$. We have already seen that the entanglement-sharing protocol immediately followed by $Z$-basis measurements of the qubits is equivalent to Steps 1 through 3 of the actual protocol. Here we consider a slightly modified scenario in which, after the entanglement-sharing protocol, a controlled-NOT operation $V$ is applied on each pair of qubits, where $V := |0\rangle\langle 0|_A \otimes \mathbf{1}_B + |1\rangle\langle 1|_A \otimes X_B$ with $X_B := |1\rangle\langle 0|_B + |0\rangle\langle 1|_B$. Alice then measures her qubits on the $Z$ basis to define her sifted key bits and proceeds with Step 5 of the actual protocol. Since $V$ does not affect the $Z$-basis value of the Alice's qubit, her procedure of determining the $\hat{N}^{\text{fin}}$-bit final key in this scenario is equivalent to that in the actual protocol. Although $V$ prevents Bob from obtaining an equivalent final key, he can still simulate the reconciliation and the verification process in Step 4 since the $Z$-basis value of each of his $\hat{N}^{\text{suc}}$ qubits corresponds to absence/presence of a bit error between Alice's and Bob's sifted key bits. Hence Bob can equivalently carry out all the announcements in Steps 4 and 5 of the actual protocol. As a result, this scenario leads to exactly the same distribution $\Pr(N^{\text{fin}})$ and the same states $\rho_{\text{AE}|N^{\text{fin}}}^{\text{fin}}$ as those of the actual protocol.

The secrecy of Alice's final key can be determined from the $X$-basis property of her $\hat{N}^{\text{suc}}$ qubits after the application of $V$. Since $V$ can be rewritten as $V = \mathbf{1}_A \otimes |+\rangle\langle +|_B + Z_A \otimes |-\rangle\langle -|_B$ with $Z_A := |-\rangle\langle +|_A + |+\rangle\langle -|_A$, the $X$-basis value of each of Alice's qubits corresponds to absence/presence of a phase error. Suppose that these $\hat{N}^{\text{suc}}$ qubits are measured on the $X$-basis to produce an outcome $\hat{\boldsymbol{x}} \in \{+, -\}^{\hat{N}^{\text{suc}}}$. Let $\text{wt}(\hat{\boldsymbol{x}})$ be the number of symbol '$-$' in $\hat{\boldsymbol{x}}$. If Eq. (15) holds in the estimation protocol, the statistics of $\hat{\boldsymbol{x}}$ should satisfy

$$
\Pr\left[\text{wt}(\hat{\boldsymbol{x}}) \leq U(\hat{F}, \hat{N}^{\text{trash}})\right] \geq 1 - \epsilon,
\tag{44}
$$

which implies that the number of probable patterns $\hat{\boldsymbol{x}}$ is limited. To be more precise, let us introduce a set $\Omega(n, w) := \{\boldsymbol{x} \in \{+, -\}^n | \text{wt}(\boldsymbol{x}) \leq w\}$, whose size is bounded as $|\Omega(n, w)| \leq 2^{nh(w/n)}$. The condition (44) then implies

$$
\Pr\left[\hat{N}^{\text{suc}} \geq 1, \hat{\boldsymbol{x}} \notin \mathcal{T}(\hat{N}^{\text{suc}}, \hat{F}, \hat{N}^{\text{trash}})\right] \leq \epsilon
\tag{45}
$$

with $\mathcal{T}(\hat{N}^{\text{suc}}, \hat{F}, \hat{N}^{\text{trash}}) := \Omega\left(\hat{N}^{\text{suc}}, U(\hat{F}, \hat{N}^{\text{trash}})\right)$, which satisfies

$$
\log_2 |\mathcal{T}(\hat{N}^{\text{suc}}, \hat{F}, \hat{N}^{\text{trash}})| \leq \hat{N}^{\text{suc}} - \hat{N}^{\text{fin}} - s
\tag{46}
$$

from Eq. (9). It is known[40,41,51] that the above conditions imply Eq. (43) with $\epsilon_{\text{sct}} = \sqrt{2}\sqrt{\epsilon + 2^{-s}}$. Therefore, it suffices to show that Eq. (15) in the estimation protocol for proving the actual protocol to be $\epsilon_{\text{sec}}$-secure with $\epsilon_{\text{sec}} = \sqrt{2}\sqrt{\epsilon + 2^{-s}} + 2^{-s'}$.

We remark that the encryption of $M := H_{\text{EC}} + s'$ bits in Step 4 can be omitted as long as each bit linearly depends on Alice's sifted key over GF(2). In such a case, the above scenario must include measurements on Alice's qubits to simulate the announcement of the $M$ bits in Step 4. The backaction on $X$ basis caused by the measurement for each bit amounts to doubling the number of probable patterns $\hat{\boldsymbol{x}}$.

We can thus redefine the set $\mathcal{T}(\hat{N}^{\mathrm{suc}}, \hat{F}, \hat{N}^{\mathrm{trash}})$ by enlarging its size by factor of $2^M$ such that Eq. (45) holds. Then, by decreasing $\hat{N}^{\mathrm{fin}}$ by $M$, Eq. (46) also holds. This means that we achieve the same net key rate with the same level of security.

*3. Derivation of the operator inequality.* The aim of this subsection is to construct $B(\kappa, \gamma)$ which fulfills the operator inequality (27). Let $\sigma_{\mathrm{sup}}(O)$ denote the supremum of the spectrum of a bounded self-adjoint operator $O$. Although $B(\kappa, \gamma) = \sigma_{\mathrm{sup}}(M[\kappa, \gamma])$ would give the tightest bound satisfying Eq. (27), it is hard to compute it numerically since system $C$ has an infinite-dimensional Hilbert space. Instead, we derive a looser but simpler bound. We first prove the following lemma.

*Lemma 1*: Let $\Pi_{\pm}$ be orthogonal projections satisfying $\Pi_+\Pi_- = 0$. Suppose that the rank of $\Pi_{\pm}$ is no smaller than 2 or infinite. Let $M_{\pm}$ be self-adjoint operators satisfying $\Pi_{\pm}M_{\pm}\Pi_{\pm} = M_{\pm} \leq \alpha_{\pm}\Pi_{\pm}$, where $\alpha_{\pm}$ are real constants. Let $|\psi\rangle$ be an unnormalized vector satisfying $(\Pi_+ + \Pi_-)|\psi\rangle = |\psi\rangle$ and $\Pi_{\pm}|\psi\rangle \neq 0$. Define following quantities with respect to $|\psi\rangle$:

$$C_{\pm} := \langle\psi|\Pi_{\pm}|\psi\rangle (>0), \tag{47}$$

$$D_{\pm} := C_{\pm}^{-1}\langle\psi|M_{\pm}|\psi\rangle, \tag{48}$$

$$V_{\pm} := C_{\pm}^{-1}\langle\psi|M_{\pm}^2|\psi\rangle - D_{\pm}^2. \tag{49}$$

Then, for any real numbers $\gamma_+$ and $\gamma_-$, we have

$$\sigma_{\mathrm{sup}}\left(M_+ + M_- + |\psi\rangle\langle\psi| - \gamma_+\Pi_+ - \gamma_-\Pi_-\right) \leq \sigma_{\mathrm{sup}}(M_{4\mathrm{d}}), \tag{50}$$

where four dimensional matrix $M_{4\mathrm{d}}$ is defined as

$$M_{4\mathrm{d}} := \begin{bmatrix} \alpha_+ - \gamma_+ & \sqrt{V_+} & 0 & 0 \\ \sqrt{V_+} & C_+ + D_+ - \gamma_+ & \sqrt{C_+ C_-} & 0 \\ 0 & \sqrt{C_+ C_-} & C_- + D_- - \gamma_- & \sqrt{V_-} \\ 0 & 0 & \sqrt{V_-} & \alpha_- - \gamma_- \end{bmatrix}. \tag{51}$$

*Proof*: We choose orthonormal vectors $\left\{\left|e_{\pm}^{(1)}\right\rangle, \left|e_{\pm}^{(2)}\right\rangle\right\}$ in the domain of $\Pi_{\pm}$, respectively, to satisfy

$$\sqrt{C_{\pm}}\left|e_{\pm}^{(1)}\right\rangle = \Pi_{\pm}|\psi\rangle, \tag{52}$$

$$M_{\pm}\left|e_{\pm}^{(1)}\right\rangle = D_{\pm}\left|e_{\pm}^{(1)}\right\rangle + \sqrt{V_{\pm}}\left|e_{\pm}^{(2)}\right\rangle, \tag{53}$$

which is well-defined due to Eqs. (47)–(49) and $\Pi_{\pm}M_{\pm}\Pi_{\pm} = M_{\pm}$. From $(\Pi_+ + \Pi_-)|\psi\rangle = |\psi\rangle$, we have

$$|\psi\rangle = \sqrt{C_+}\left|e_+^{(1)}\right\rangle + \sqrt{C_-}\left|e_-^{(1)}\right\rangle. \tag{54}$$

Let us define the following projection operators:

$$\Pi_{\pm}^{(j)} := \left|e_{\pm}^{(j)}\right\rangle\left\langle e_{\pm}^{(j)}\right| \quad (j = 1, 2), \tag{55}$$

$$\Pi_{\pm}^{(\geq 2)} := \Pi_{\pm} - \Pi_{\pm}^{(1)}, \tag{56}$$

$$\Pi_{\pm}^{(\geq 3)} := \Pi_{\pm}^{(\geq 2)} - \Pi_{\pm}^{(2)}. \tag{57}$$

Since Eq. (53) implies $\Pi_{\pm}^{(\geq 3)}M_{\pm}\Pi_{\pm}^{(1)} = 0$, we have

$$M_{\pm} = \Pi_{\pm}^{(1)}M_{\pm}\Pi_{\pm}^{(1)} + \Pi_{\pm}^{(\geq 2)}M_{\pm}\Pi_{\pm}^{(\geq 2)} + \Pi_{\pm}^{(1)}M_{\pm}\Pi_{\pm}^{(2)} + \Pi_{\pm}^{(2)}M_{\pm}\Pi_{\pm}^{(1)}. \tag{58}$$

The second term in the right-hand side of Eq. (58) is bounded as

$$\Pi_{\pm}^{(\geq 2)}M_{\pm}\Pi_{\pm}^{(\geq 2)} \leq \alpha_{\pm}\Pi_{\pm}^{(\geq 2)}, \tag{59}$$

since $M_{\pm} \leq \alpha_{\pm}\Pi_{\pm}$. Combining Eqs. (48), (58), and (59), we have

$$M_{\pm} - \gamma_{\pm}\Pi_{\pm}$$
$$\leq (D_{\pm} - \gamma_{\pm})\left|e_{\pm}^{(1)}\right\rangle\left\langle e_{\pm}^{(1)}\right| + (\alpha_{\pm} - \gamma_{\pm})\left|e_{\pm}^{(2)}\right\rangle\left\langle e_{\pm}^{(2)}\right|$$
$$+ \sqrt{V_{\pm}}\left(\left|e_{\pm}^{(1)}\right\rangle\left\langle e_{\pm}^{(2)}\right| + \left|e_{\pm}^{(2)}\right\rangle\left\langle e_{\pm}^{(1)}\right|\right) + (\alpha_{\pm} - \gamma_{\pm})\Pi_{\pm}^{(\geq 3)}. \tag{60}$$

Combining Eqs. (54) and (60), we have

$$M_+ + M_- + |\psi\rangle\langle\psi| - \gamma_+\Pi_+ - \gamma_-\Pi_-$$
$$\leq M_{4\mathrm{d}} \oplus (\alpha_+ - \gamma_+)\Pi_+^{(\geq 3)} \oplus (\alpha_- - \gamma_-)\Pi_-^{(\geq 3)}, \tag{61}$$

where $M_{4\mathrm{d}}$ is given in Eq. (51) with the basis $\left\{\left|e_+^{(2)}\right\rangle, \left|e_+^{(1)}\right\rangle, \left|e_-^{(1)}\right\rangle, \left|e_-^{(2)}\right\rangle\right\}$. Since $\alpha_{\pm} - \gamma_{\pm} = \left\langle e_{\pm}^{(2)}\right|M_{4\mathrm{d}}\left|e_{\pm}^{(2)}\right\rangle \leq \sigma_{\mathrm{sup}}(M_{4\mathrm{d}})$, supremum of the spectrum of the right-hand side of Eq. (61) is equal to the maximum eigenvalue of the four-dimensional matrix $M_{4\mathrm{d}}$. We thus obtain Eq. (50).

As a corollary, we derive Eq. (27) as follows.

*Corollary 2*: Let $|\beta\rangle$ be a coherent state. Let $\Pi_{\mathrm{ev(od)}}$, $M_{\mathrm{ev(od)}}^{\mathrm{suc}}$, and $M[\kappa, \gamma]$ be as defined in the main text, and define following quantities:

$$C_{\mathrm{ev}} := \langle\beta|\Pi_{\mathrm{ev}}|\beta\rangle = e^{-|\beta|^2}\cosh|\beta|^2, \tag{62}$$

$$C_{\mathrm{od}} := \langle\beta|\Pi_{\mathrm{od}}|\beta\rangle = e^{-|\beta|^2}\sinh|\beta|^2, \tag{63}$$

$$D_{\mathrm{ev(od)}} := C_{\mathrm{ev(od)}}^{-1}\langle\beta|M_{\mathrm{ev(od)}}^{\mathrm{suc}}|\beta\rangle, \tag{64}$$

$$V_{\mathrm{ev(od)}} := C_{\mathrm{ev(od)}}^{-1}\langle\beta|\left(M_{\mathrm{ev(od)}}^{\mathrm{suc}}\right)^2|\beta\rangle - D_{\mathrm{ev(od)}}^2. \tag{65}$$

Let $M_{4\mathrm{d}}^{\mathrm{err}}[\kappa, \gamma]$ and $M_{2\mathrm{d}}^{\mathrm{cor}}[\kappa, \gamma]$ be defined as follows:

$$M_{4\mathrm{d}}^{\mathrm{err}}[\kappa, \gamma] := \begin{bmatrix} 1 & \sqrt{V_{\mathrm{od}}} & & \\ \sqrt{V_{\mathrm{od}}} & \kappa\, C_{\mathrm{od}} + D_{\mathrm{od}} & \kappa\sqrt{C_{\mathrm{od}}\, C_{\mathrm{ev}}} & \\ & \kappa\sqrt{C_{\mathrm{od}}\, C_{\mathrm{ev}}}, & \kappa\, C_{\mathrm{ev}} + D_{\mathrm{ev}} - \gamma & \sqrt{V_{\mathrm{ev}}} \\ & & \sqrt{V_{\mathrm{ev}}} & 1 - \gamma \end{bmatrix}, \tag{66}$$

$$M_{2\mathrm{d}}^{\mathrm{cor}}[\kappa, \gamma] := \begin{bmatrix} \kappa\, C_{\mathrm{ev}} & \kappa\sqrt{C_{\mathrm{ev}}\, C_{\mathrm{od}}} \\ \kappa\sqrt{C_{\mathrm{ev}}\, C_{\mathrm{od}}} & \kappa\, C_{\mathrm{od}} - \gamma \end{bmatrix}. \tag{67}$$

Define a convex function

$$B(\kappa, \gamma) := \max\left\{\sigma_{\mathrm{sup}}\left(M_{4\mathrm{d}}^{\mathrm{err}}[\kappa, \gamma]\right), \sigma_{\mathrm{sup}}\left(M_{2\mathrm{d}}^{\mathrm{cor}}[\kappa, \gamma]\right)\right\}. \tag{68}$$

Then, for $\kappa, \gamma \geq 0$, we have

$$M[\kappa, \gamma] \leq B(\kappa, \gamma)\mathbf{I}_{AC}. \tag{69}$$

*Proof*: Let us first observe that the operator $\Pi^{\mathrm{fid}}$ defined in Eq. (20) can be rewritten as follows:

$$\Pi^{\mathrm{fid}} = |\phi_{\mathrm{err}}\rangle\langle\phi_{\mathrm{err}}|_{AC} + |\phi_{\mathrm{cor}}\rangle\langle\phi_{\mathrm{cor}}|_{AC}, \tag{70}$$

where orthogonal states $|\phi_{\mathrm{err}}\rangle_{AC}$ and $|\phi_{\mathrm{cor}}\rangle_{AC}$ are defined as

$$|\phi_{\mathrm{err}}\rangle_{AC} := |+\rangle_A \otimes \Pi_{\mathrm{od}}|\beta\rangle_C + |-\rangle_A \otimes \Pi_{\mathrm{ev}}|\beta\rangle_C, \tag{71}$$

$$|\phi_{\mathrm{cor}}\rangle_{AC} := |+\rangle_A \otimes \Pi_{\mathrm{ev}}|\beta\rangle_C + |-\rangle_A \otimes \Pi_{\mathrm{od}}|\beta\rangle_C. \tag{72}$$

Next, using Eqs. (70), (19), and (21), we rearrange the operator $M[\kappa, \gamma]$ defined in Eq. (26) as follows:

$$M[\kappa, \gamma] = M^{\mathrm{err}}[\kappa, \gamma] \oplus M^{\mathrm{cor}}[\kappa, \gamma], \tag{73}$$

where

$$M^{\mathrm{err}}[\kappa, \gamma] := |+\rangle\langle+|_A \otimes M_{\mathrm{od}}^{\mathrm{suc}} + |-\rangle\langle-|_A \otimes M_{\mathrm{ev}}^{\mathrm{suc}}$$
$$+ \kappa|\phi_{\mathrm{err}}\rangle\langle\phi_{\mathrm{err}}|_{AC} - \gamma|-\rangle\langle-|_A \otimes \Pi_{\mathrm{ev}}, \tag{74}$$

$$M^{\mathrm{cor}}[\kappa, \gamma] := \kappa|\phi_{\mathrm{cor}}\rangle\langle\phi_{\mathrm{cor}}|_{AC} - \gamma|-\rangle\langle-|_A \otimes \Pi_{\mathrm{od}}. \tag{75}$$

We can apply Lemma 1 to $M^{\mathrm{err}}[\kappa, \gamma]$ by the following substitutions

$$M_{\pm} = |\pm\rangle\langle\pm|_A \otimes M_{\mathrm{od(ev)}}^{\mathrm{suc}}, \tag{76}$$

$$|\psi\rangle = \sqrt{\kappa}|\phi_{\mathrm{err}}\rangle_{AC}, \tag{77}$$

$$\Pi_{\pm} = |\pm\rangle\langle\pm|_A \otimes \Pi_{\mathrm{od(ev)}}, \tag{78}$$

$$\alpha_{\pm} = 1, \tag{79}$$

$$\gamma^+ = 0, \quad \gamma^- = \gamma. \tag{80}$$

Here, $M_{\pm} \leq \Pi_{\pm}$ (i.e., $\alpha_{\pm} = 1$) holds because $M_{\mathrm{od(ev)}}^{\mathrm{suc}}$ are POVM elements. The other prerequisites of Lemma 1 are easy to be confirmed. Thus, we obtain

$$\sigma_{\mathrm{sup}}(M^{\mathrm{err}}[\kappa, \gamma]) \leq \sigma_{\mathrm{sup}}\left(M_{4\mathrm{d}}^{\mathrm{err}}[\kappa, \gamma]\right). \tag{81}$$

In the same way, we can apply Lemma 1 to $M^{\mathrm{cor}}[\kappa, \gamma]$ via

$$M_{\pm} = 0, \tag{82}$$

$$|\psi\rangle = \sqrt{\kappa}|\phi_{\mathrm{cor}}\rangle_{AC}, \tag{83}$$

$$\Pi_{\pm} = |\pm\rangle\langle\pm|_A \otimes \Pi_{\mathrm{ev(od)}}, \tag{84}$$

$$\alpha_{\pm} = 0, \tag{85}$$

$$\gamma^+ = 0, \quad \gamma^- = \gamma. \tag{86}$$

Since $M_{\pm} = 0$ implies $D_{\pm} = V_{\pm} = 0$ in Lemma 1, this time we can reduce the dimension of relevant matrix Eq. (51) by separating known eigenvalues 0 and $-\gamma$. Therefore, we have

$$\sigma_{\sup}(M^{\mathrm{cor}}[\kappa, \gamma]) \le \max\left\{\sigma_{\sup}\left(M_{2\mathrm{d}}^{\mathrm{cor}}[\kappa, \gamma]\right), 0, -\gamma\right\} \tag{87}$$
$$= \sigma_{\sup}\left(M_{2\mathrm{d}}^{\mathrm{cor}}[\kappa, \gamma]\right),$$

where the last inequality holds since $\gamma \ge 0$ and $\kappa C_{\mathrm{ev}} \ge 0$. We then obtain Eq. (69) from Eqs. (73), (81), and (87). Since $M_{4\mathrm{d}}^{\mathrm{err}}[\kappa, \gamma]$ and $M_{2\mathrm{d}}^{\mathrm{cor}}[\kappa, \gamma]$ are symmetric and their elements linearly depend on $\kappa$ and $\gamma$, $\sigma_{\sup}\left(M_{4\mathrm{d}}^{\mathrm{err}}[\kappa, \gamma]\right)$ and $\sigma_{\sup}\left(M_{2\mathrm{d}}^{\mathrm{cor}}[\kappa, \gamma]\right)$ are convex functions over $\kappa$ and $\gamma$, and so is $B(\kappa, \gamma)$.

*4. Derivation of the finite-size bound.* Here we construct the function $U(\hat{F}, \hat{N}^{\mathrm{trash}})$ to satisfy Eq. (15) in the estimation protocol. For that, we will first derive Eq. (30). In the estimation protocol, we define the following random variables labeled by the number $i$ of the round;

(i) $\hat{N}_{\mathrm{ph}}^{\mathrm{suc},(i)}$ is defined to be unity only when "signal" is chosen in the $i$-th round, the detection is a "success", and a pair of outcomes $(a', b')$ is $(+, -)$ or $(-, +)$. Otherwise, $\hat{N}_{\mathrm{ph}}^{\mathrm{suc},(i)} = 0$. We have

$$\hat{N}_{\mathrm{ph}}^{\mathrm{suc},(i)} = \begin{cases} 1 & (\text{signal, success}, (+, -) \text{ or } (-, +)) \\ 0 & (\text{otherwise}), \end{cases} \tag{88}$$

and $\hat{N}_{\mathrm{ph}}^{\mathrm{suc}} = \sum_{i=1}^{N} \hat{N}_{\mathrm{ph}}^{\mathrm{suc},(i)}$.

(ii) $\hat{F}^{(i)}$ is defined to be $\Lambda_{m,r}(|\hat{\omega} - (-1)^a \beta|^2)$ when "test" is chosen in the $i$-th round. We have

$$\hat{F}^{(i)} = \begin{cases} \Lambda_{m,r}(|\hat{\omega} - (-1)^a \beta|^2) & (\text{test}) \\ 0 & (\text{otherwise}), \end{cases} \tag{89}$$

and $\hat{F} = \sum_{i=1}^{N} \hat{F}^{(i)}$.

(iii) $\hat{Q}_-^{(i)}$ is defined to be unity only when "trash" is chosen in the $i$-th round and $a' = -$. Otherwise, $\hat{Q}_-^{(i)} = 0$. We have

$$\hat{Q}_-^{(i)} = \begin{cases} 1 & (\text{trash}, -) \\ 0 & (\text{otherwise}), \end{cases} \tag{90}$$

and $\hat{Q}_- = \sum_{i=1}^{N} \hat{Q}_-^{(i)}$.

(iv) We also define

$$\hat{T}^{(i)} := p_{\mathrm{sig}}^{-1} \hat{N}_{\mathrm{ph}}^{\mathrm{suc},(i)} + p_{\mathrm{test}}^{-1} \kappa \hat{F}^{(i)} - p_{\mathrm{trash}}^{-1} \gamma \hat{Q}_-^{(i)}, \tag{91}$$

which leads to $\hat{T}[\kappa, \gamma] = \sum_{i=1}^{N} \hat{T}^{(i)}$.

We will make use of Azuma's inequality[42]. We define stochastic processes $\{\hat{X}^{(k)}\}_{k=0,\dots,N}$ and $\{\hat{Y}^{(k)}\}_{k=1,\dots,N}$ as follows:

$$\hat{X}^{(0)} := 0, \tag{92}$$

$$\hat{X}^{(k)} := \sum_{i=1}^{k} \left(\hat{T}^{(i)} - \hat{Y}^{(i)}\right) \quad (k \ge 1), \tag{93}$$

$$\hat{Y}^{(k)} := \mathbb{E}\left[\hat{T}^{(k)} \middle| \hat{X}^{<k}\right], \tag{94}$$

where $\hat{X}^{<k} := (\hat{X}^{(0)}, \hat{X}^{(1)}, \dots, \hat{X}^{(k-1)})$. Note that $\hat{Y}^{(k)}$ is a constant when conditioned on $\hat{X}^{<k}$. Such a sequence $\{\hat{Y}^{(k)}\}_{k=1,2,\dots}$ is called a predictable process with regards to $\{\hat{X}^{(k)}\}$. Since $\hat{T}^{(i)}$ is bounded for any $i$ and $\{\hat{X}^{(k)}\}_{k=0,1,\dots}$ is a martingale, we can apply Azuma's inequality.

*Proposition 1* (Azuma's inequality[52,53]): Suppose $\{\hat{X}^{(k)}\}_{k=0,1,\dots}$ is a martingale which satisfies

$$-\hat{Y}^{(k)} + c_{\min} \le \hat{X}^{(k)} - \hat{X}^{(k-1)} \le -\hat{Y}^{(k)} + c_{\max}, \tag{95}$$

for constants $c_{\min}$ and $c_{\max}$, and a predictable process $\{\hat{Y}^{(k)}\}_{k=1,2,\dots}$ with regards to $\{\hat{X}^{(k)}\}$, i.e., $\hat{Y}^{(k)}$ is constant when conditioned on $\hat{X}^{<k}$. Then, for all positive integers $N$ and all positive reals $\delta$,

$$\Pr[\hat{X}^{(N)} - \hat{X}^{(0)} \ge \delta] \le \exp\left(-\frac{2\delta^2}{(c_{\max} - c_{\min})^2 N}\right). \tag{96}$$

We define constants $c_{\min}$ and $c_{\max}$ as follows. In each round, at most one of $\hat{N}_{\mathrm{ph}}^{\mathrm{suc},(i)}$, $\hat{F}^{(i)}$, and $\hat{Q}_-^{(i)}$ takes non-zero value; $\hat{N}_{\mathrm{ph}}^{\mathrm{suc},(i)}$ and $\hat{Q}_-^{(i)}$ are either zero or unity, and $\min \Lambda_{m,r} \le \hat{F}^{(i)} \le \max \Lambda_{m,r}$. Since $\kappa, \gamma \ge 0$, Eq. (95) holds when $c_{\min}$ and $c_{\max}$ are defined as

$$c_{\min} := \min\left(p_{\mathrm{test}}^{-1} \kappa \ \min \Lambda_{m,r}, -p_{\mathrm{trash}}^{-1} \gamma, 0\right), \tag{97}$$

$$c_{\max} := \max\left(p_{\mathrm{sig}}^{-1}, p_{\mathrm{test}}^{-1} \kappa \ \max \Lambda_{m,r}, 0\right). \tag{98}$$

With $c_{\min}$ and $c_{\max}$ defined as above, we further define

$$\delta_1(\epsilon) := (c_{\max} - c_{\min}) \sqrt{\frac{N}{2} \ln\left(\frac{1}{\epsilon}\right)}. \tag{99}$$

Setting $\delta = \delta_1(\epsilon/2)$ in the proposition, we conclude that

$$\hat{T}[\kappa, \gamma] \le \sum_{i=1}^{N} \hat{Y}^{(i)} + \delta_1(\epsilon/2) \tag{100}$$

holds with a probability no smaller than $1 - \epsilon/2$.

Next, we will construct a deterministic bound on $\hat{Y}^{(i)}$. Let $\rho_{AC}^{(i)}$ be the state of Alice's $i$-th qubit and Bob's $i$-th pulse conditioned on $\hat{X}^{<i}$. Then, using the same argument as that has lead to Eqs. (22)–(24), we have

$$\mathbb{E}\left[\hat{N}_{\mathrm{ph}}^{\mathrm{suc},(i)} \middle| \hat{X}^{<i}\right] = p_{\mathrm{sig}} \mathrm{Tr}\left(\rho_{AC}^{(i)} M_{\mathrm{ph}}^{\mathrm{suc}}\right), \tag{101}$$

$$\mathbb{E}\left[\hat{Q}_-^{(i)} \middle| \hat{X}^{<i}\right] = p_{\mathrm{trash}} \mathrm{Tr}\left(\rho_{AC}^{(i)} \Pi_-^{\mathrm{trash}}\right), \tag{102}$$

$$\mathbb{E}\left[\hat{F}^{(i)} \middle| \hat{X}^{<i}\right] \le p_{\mathrm{fid}} \mathrm{Tr}\left(\rho_{AC}^{(i)} \Pi^{\mathrm{fid}}\right), \tag{103}$$

and thus

$$\hat{Y}^{(i)} \le \mathrm{Tr}\left(\rho_{AC}^{(i)} M[\kappa, \gamma]\right), \tag{104}$$

where $M[\kappa, \gamma]$ is defined in Eq. (26). Using the operator inequality (27), we obtain a bound independent of $i$ as

$$\hat{Y}^{(i)} \le B(\kappa, \gamma). \tag{105}$$

Combining this with Eq. (100) proves Eq. (30).

The function $\delta_2(\epsilon/2; \hat{N}^{\mathrm{trash}})$ satisfying the bound (31) on $\hat{Q}_-$ can be derived from the fact that $\Pr[\hat{Q}_- | \hat{N}^{\mathrm{trash}}]$ is a binomial distribution. The following inequality thus holds for any positive integer $n$ and a real $\delta$ with $0 < \delta < (1 - q_-)n$ (Chernoff bound):

$$\Pr\left[\hat{Q}_- - q_- n \ge \delta \middle| \hat{N}^{\mathrm{trash}} = n\right] \le 2^{-nD(q_- + \delta/n \| q_-)}, \tag{106}$$

where

$$D(x \| y) := x \log_2 \frac{x}{y} + (1 - x) \log_2 \frac{1 - x}{1 - y} \tag{107}$$

is the Kullback-Leibler divergence. On the other hand, for any non-negative integer $n$, we always have

$$\Pr\left[\hat{Q}_- - q_- n \le (1 - q_-)n \middle| \hat{N}^{\mathrm{trash}} = n\right] = 1. \tag{108}$$

Therefore, for any non-negative integer $n$, by defining $\delta_2(\epsilon; n)$ which satisfies

$$\begin{cases} D(q_- + \delta_2(\epsilon; n)/n \| q_-) = -\frac{1}{n} \log_2(\epsilon) & (\epsilon > q_-^n) \\ \delta_2(\epsilon; n) = (1 - q_-)n & (\epsilon \le q_-^n) \end{cases}, \tag{109}$$

and by combining Eq. (106) and (108), we conclude that Eq. (31) holds with a probability no smaller than $1 - \epsilon/2$.

Combining Eq. (30) and Eq. (31), we obtain Eq. (15) by setting

$$U(\hat{F}, \hat{N}^{\mathrm{trash}}) := p_{\mathrm{sig}} NB(\kappa, \gamma) + p_{\mathrm{sig}} \delta_1(\epsilon/2) \\ - \frac{p_{\mathrm{sig}}}{p_{\mathrm{test}}} \kappa \hat{F} + \frac{p_{\mathrm{sig}}}{p_{\mathrm{trash}}} \gamma \left(q_- \hat{N}^{\mathrm{trash}} + \delta_2(\epsilon/2; \hat{N}^{\mathrm{trash}})\right), \tag{110}$$

which holds with a probability no smaller than $1 - \epsilon$ (Union bound). Note that since $B(\kappa, \gamma)$ is a convex function, so is $U(\hat{F}, \hat{N}^{\mathrm{trash}})$ with respect to auxiliary parameters $\kappa$ and $\gamma$.

**Models for numerical simulation of key rates.** In what follows, we normalize quadrature $x$ such that a coherent state $|\omega\rangle$ has expectation $\langle x \rangle = \mathrm{Re}(\omega)$ and variance $\langle (\Delta x)^2 \rangle = 1/4$. The wave function for $\omega = \omega_R + i\omega_I$ is given by

$$\langle x | \omega \rangle = \left(\frac{2}{\pi}\right)^{\frac{1}{4}} \exp\left[-(x - \omega_R)^2 + 2i\omega_I x - i\omega_R \omega_I\right]. \tag{111}$$

For the simulation of the key rate $\hat{G}$, we assume that the communication channel and Bob's detection apparatus can be modeled by a pure loss channel followed by random displacement, that is, the states which Bob receives are given by

$$\rho_{\mathrm{model}}^{(a)} := \int_{\mathbb{C}} p_\xi(\gamma) |(-1)^a \sqrt{\eta\mu} + \gamma\rangle \langle (-1)^a \sqrt{\eta\mu} + \gamma| d^2\gamma, \tag{112}$$

**Table 1 Examples of optimized parameters.**

| $\eta$ [dB] | Key rate $\hat{G}$ | $(\kappa, \gamma)$ | $\mu$ | $x_{\text{th}}$ | $p_{\text{sig}}$ | $p_{\text{test}}$ |
|---|---|---|---|---|---|---|
| Parameters for $N = 10^{11}$ and $\xi = 0$ | | | | | | |
| 0.5 | $2.17 \times 10^{-1}$ | (44.9, 1.92) | 0.554 | 0.451 | 0.821 | 0.172 |
| 1.0 | $1.38 \times 10^{-1}$ | (32.4, 1.38) | 0.514 | 0.532 | 0.821 | 0.172 |
| 1.5 | $8.71 \times 10^{-2}$ | (24.9, 1.01) | 0.487 | 0.610 | 0.816 | 0.176 |
| 2.0 | $5.36 \times 10^{-2}$ | (20.3, 0.741) | 0.442 | 0.724 | 0.831 | 0.160 |
| 2.5 | $3.16 \times 10^{-2}$ | (17.0, 0.538) | 0.459 | 0.771 | 0.788 | 0.205 |
| 3.0 | $1.75 \times 10^{-2}$ | (14.6, 0.381) | 0.451 | 0.855 | 0.767 | 0.227 |
| 3.5 | $8.92 \times 10^{-3}$ | (13.6, 0.262) | 0.446 | 0.941 | 0.706 | 0.289 |
| 4.0 | $4.15 \times 10^{-3}$ | (12.6, 0.175) | 0.442 | 1.03 | 0.624 | 0.371 |
| 4.5 | $1.50 \times 10^{-3}$ | (11.4, 0.107) | 0.439 | 1.13 | 0.522 | 0.473 |
| 5.0 | $3.23 \times 10^{-4}$ | (9.98, 0.059) | 0.438 | 1.25 | 0.370 | 0.626 |
| 5.5 | $1.63 \times 10^{-5}$ | (8.78, 0.031) | 0.443 | 1.37 | 0.126 | 0.869 |
| Parameters for $N = 10^{11}$ and $\xi = 10^{-3.0}$ | | | | | | |
| 0.5 | $1.79 \times 10^{-1}$ | (23.5, 1.60) | 0.491 | 0.489 | 0.854 | 0.137 |
| 1.0 | $1.13 \times 10^{-1}$ | (17.9, 1.19) | 0.466 | 0.567 | 0.853 | 0.138 |
| 1.5 | $6.95 \times 10^{-2}$ | (14.3, 0.889) | 0.450 | 0.645 | 0.848 | 0.143 |
| 2.0 | $4.07 \times 10^{-2}$ | (11.9, 0.666) | 0.442 | 0.724 | 0.831 | 0.160 |
| 2.5 | $2.20 \times 10^{-2}$ | (10.1, 0.487) | 0.439 | 0.808 | 0.804 | 0.187 |
| 3.0 | $1.02 \times 10^{-2}$ | (8.85, 0.345) | 0.440 | 0.898 | 0.758 | 0.233 |
| 3.5 | $3.56 \times 10^{-3}$ | (7.72, 0.232) | 0.447 | 0.999 | 0.674 | 0.316 |
| 4.0 | $5.29 \times 10^{-4}$ | (6.70, 0.147) | 0.463 | 1.11 | 0.484 | 0.505 |
| Parameters for $N = 10^{12}$ and $\xi = 0$ | | | | | | |
| 0.5 | $2.68 \times 10^{-1}$ | (82.9, 2.26) | 0.616 | 0.421 | 0.864 | 0.133 |
| 1.0 | $1.69 \times 10^{-1}$ | (54.9, 1.56) | 0.556 | 0.504 | 0.874 | 0.123 |
| 1.5 | $1.08 \times 10^{-1}$ | (40.3, 1.11) | 0.518 | 0.590 | 0.873 | 0.124 |
| 2.0 | $6.76 \times 10^{-2}$ | (31.3, 0.798) | 0.493 | 0.670 | 0.868 | 0.128 |
| 2.5 | $4.12 \times 10^{-2}$ | (26.5, 0.575) | 0.475 | 0.750 | 0.852 | 0.145 |
| 3.0 | $2.41 \times 10^{-2}$ | (23.2, 0.408) | 0.466 | 0.834 | 0.829 | 0.168 |
| 3.5 | $1.32 \times 10^{-2}$ | (20.1, 0.275) | 0.449 | 0.919 | 0.803 | 0.195 |
| 4.0 | $6.93 \times 10^{-3}$ | (17.7, 0.184) | 0.443 | 1.01 | 0.772 | 0.226 |
| 4.5 | $3.15 \times 10^{-3}$ | (16.4, 0.115) | 0.434 | 1.10 | 0.698 | 0.300 |
| 5.0 | $1.14 \times 10^{-3}$ | (14.6, 0.065) | 0.427 | 1.21 | 0.596 | 0.402 |
| 5.5 | $3.29 \times 10^{-4}$ | (12.9, 0.036) | 0.423 | 1.32 | 0.467 | 0.531 |
| 6.0 | $3.23 \times 10^{-5}$ | (10.8, 0.017) | 0.421 | 1.45 | 0.240 | 0.759 |
| Parameters for $N = 10^{12}$ and $\xi = 10^{-3.0}$ | | | | | | |
| 0.5 | $2.09 \times 10^{-1}$ | (29.4, 1.71) | 0.513 | 0.474 | 0.906 | 0.089 |
| 1.0 | $1.32 \times 10^{-1}$ | (21.7, 1.25) | 0.482 | 0.554 | 0.909 | 0.086 |
| 1.5 | $8.23 \times 10^{-2}$ | (16.9, 0.928) | 0.462 | 0.633 | 0.909 | 0.086 |
| 2.0 | $4.92 \times 10^{-2}$ | (13.8, 0.689) | 0.450 | 0.712 | 0.899 | 0.096 |
| 2.5 | $2.74 \times 10^{-2}$ | (11.5, 0.502) | 0.444 | 0.797 | 0.888 | 0.107 |
| 3.0 | $1.36 \times 10^{-2}$ | (9.82, 0.355) | 0.442 | 0.886 | 0.858 | 0.137 |
| 3.5 | $5.28 \times 10^{-3}$ | (8.26, 0.237) | 0.446 | 0.989 | 0.834 | 0.161 |
| 4.0 | $1.17 \times 10^{-3}$ | (7.13, 0.151) | 0.458 | 1.10 | 0.701 | 0.293 |

Examples of parameters for a given pair of total number of rounds $N$ and an excess noise parameter $\xi$ defined in Eq. (114). Given $(N, \xi)$, protocol parameters $(\kappa, \gamma, \mu, x_{\text{th}}, p_{\text{sig}}, p_{\text{test}})$ are optimized for each attenuation $\eta$ [dB] so that the net key gain per pulse (key rate) $\hat{G}$ is maximized.

where $\eta$ is the transmissivity of the pure loss channel and $p_\xi(\gamma)$ is given by

$$p_\xi(\gamma) := \frac{2}{\pi \xi} e^{-2|\gamma|^2/\xi}. \tag{113}$$

The parameter $\xi$ is the excess noise relative to the vacuum, namely,

$$\langle (\Delta x)^2 \rangle_{\rho_{\text{model}}^{(a)}} = (1 + \xi)/4. \tag{114}$$

We assume that Bob sets $\beta = \sqrt{\eta\mu}$ for the fidelity test. The actual fidelity between Bob's objective state $\left| (-1)^a \sqrt{\eta\mu} \right\rangle$ and the model state $\rho_{\text{model}}^{(a)}$ is given by

$$\begin{aligned} &F(\rho_{\text{model}}^{(a)}, \left| (-1)^a \sqrt{\eta\mu} \right\rangle \left\langle (-1)^a \sqrt{\eta\mu} \right|) \\ &= \int_{\mathbb{C}} p_\xi(\gamma) |\langle (-1)^a \sqrt{\eta\mu} | (-1)^a \sqrt{\eta\mu} - \gamma \rangle|^2 d^2\gamma \\ &= \frac{1}{1 + \xi/2}. \end{aligned} \tag{115}$$

For the acceptance probability of Bob's measurement in the signal rounds, we assume $f_{\text{suc}}(x) = \Theta(|x| - x_{\text{th}})$, a step function with the threshold $x_{\text{th}} > 0$. In this case, the quantities defined in Eqs. (64) and (65) are given by

$$D_{\text{ev}} = \int_0^\infty 2C_{\text{ev}}^{-1} f_{\text{suc}}(x) \, |\langle x|\Pi_{\text{ev}}|\beta\rangle|^2 dx \tag{116}$$

$$\begin{aligned} &= \frac{1}{4C_{\text{ev}}} \left[ \text{erfc}\left( \sqrt{2}(x_{\text{th}} - \beta) \right) + \text{erfc}\left( \sqrt{2}(x_{\text{th}} + \beta) \right) \right. \\ &\left. + 2e^{-2\beta^2} \text{erfc}\left( \sqrt{2}x_{\text{th}} \right) \right], \end{aligned} \tag{117}$$

$$D_{\text{od}} = \int_0^\infty 2C_{\text{od}}^{-1} f_{\text{suc}}(x) \, |\langle x|\Pi_{\text{od}}|\beta\rangle|^2 dx \tag{118}$$

$$\begin{aligned} &= \frac{1}{4C_{\text{od}}} \left[ \text{erfc}\left( \sqrt{2}(x_{\text{th}} - \beta) \right) + \text{erfc}\left( \sqrt{2}(x_{\text{th}} + \beta) \right) \right. \\ &\left. - 2e^{-2\beta^2} \text{erfc}\left( \sqrt{2}x_{\text{th}} \right) \right], \end{aligned} \tag{119}$$

$$V_{\text{ev(od)}} = \int_0^\infty 2C_{\text{ev(od)}}^{-1} \left( f_{\text{suc}}(x) \right)^2 \left| \langle x|\Pi_{\text{ev(od)}}|\beta\rangle \right|^2 dx - D_{\text{ev(od)}}^2 \tag{120}$$

$$= D_{\text{ev(od)}} - D_{\text{ev(od)}}^2, \tag{121}$$

where $\beta = \sqrt{\eta\mu}$ and the complementary error function $\text{erfc}(x)$ is defined as

$$\mathrm{erfc}(x) := \frac{2}{\sqrt{\pi}} \int_x^\infty dt\, e^{-t^2}. \tag{122}$$

For the derivation of Eq. (120), we used the fact that $\Pi_{\mathrm{ev}} + \Pi_{\mathrm{od}} = \mathbf{1}$ and $(\Pi_{\mathrm{ev}} - \Pi_{\mathrm{od}})|\beta\rangle = |-\beta\rangle$.

We assume that the number of "success" signal rounds $\hat{N}^{\mathrm{suc}}$ is equal to its expectation value,

$$\mathbb{E}[\hat{N}^{\mathrm{suc}}] = \left( \int_{-\infty}^\infty f(|x|)\langle x|\rho_{\mathrm{model}}^{(a)}|x\rangle dx \right) p_{\mathrm{sig}} N \tag{123}$$
$$= p_{\mathrm{sig}} N (P^+ + P^-),$$

where

$$P^\pm := \int_{x_{\mathrm{th}}}^\infty \langle \pm(-1)^a x | \rho_{\mathrm{model}}^{(a)} | \pm(-1)^a x\rangle dx$$
$$= \frac{1}{2}\, \mathrm{erfc}\left( (x_{\mathrm{th}} \mp \sqrt{\eta\mu}) \sqrt{\frac{2}{1+\xi}} \right). \tag{124}$$

We also assume that the number of test rounds $\hat{N}^{\mathrm{test}}$ is equal to $p_{\mathrm{test}} N$ and the number of trash rounds $\hat{N}^{\mathrm{trash}}$ is equal to $p_{\mathrm{trash}} N$. The test outcome $\hat{F}$ is assumed to be equal to its expectation value $\mathbb{E}[\hat{F}]$, which is given by

$$\mathbb{E}[\hat{F}] = p_{\mathrm{test}} N\, \mathbb{E}_{\rho_{\mathrm{model}}^{(a)}}[\Lambda_{m,r}(|\hat{\omega} - (-1)^a\sqrt{\eta\mu}|^2)]$$
$$= p_{\mathrm{test}} N \int_{\mathbb{C}} \frac{d^2\omega}{\pi} \langle \omega | \rho_{\mathrm{model}}^{(a)} | \omega \rangle \Lambda_{m,r}(|\omega - (-1)^a\sqrt{\eta\mu}|^2) \tag{125}$$
$$= \frac{p_{\mathrm{test}} N}{1+\xi/2}\left[ 1 - (-1)^{m+1}\left( \frac{\xi/2}{1 + r(1+\xi/2)} \right)^{m+1} \right].$$

Under these assumptions, the key rate $\hat{G}$ for each transmissivity $\eta$ is optimized over two coefficients ($\kappa$, $\gamma$) and four protocol parameters ($\mu$, $x_{\mathrm{th}}$, $p_{\mathrm{sig}}$, $p_{\mathrm{test}}$) as discussed in the main part. Examples of optimized parameters are shown in Table 1. The cost of bit error correction $H_{\mathrm{EC}}$ is assumed to be $1.1 \times \hat{N}^{\mathrm{suc}} h(e_{\mathrm{bit}})$, where the bit error rate $e_{\mathrm{bit}}$ is given by

$$e_{\mathrm{bit}} = \frac{P^-}{P^+ + P^-}. \tag{126}$$

## Data availability
Data sharing not applicable to the article as no datasets were generated or analyzed during the current study.

## Code availability
Computer codes to calculate the key rates are available from the corresponding author upon reasonable request.

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

## Acknowledgements

This work was supported by Cross-ministerial Strategic Innovation Promotion Program (SIP) (Council for Science, Technology and Innovation (CSTI)); CREST (Japan Science and Technology Agency) JPMJCR1671; JSPS KAKENHI Grant Number JP18K13469.

## Author contributions

T.M., K.M., T.S. and M.K. contributed to the initial conception of the ideas, to the working out of details, and to the writing and editing of the manuscript.

## Competing interests

The authors declare no competing interests.
