## [Peer Review File · Nature Communications]

Reviewer #1 (Remarks to the Author):

This paper addresses a major open problem in the field of quantum key distribution (qkd) with continuous variables. While these protocols present many experimental advantages, it is difficult to establish their security with the same level of generality than for qubit-based protocols such as BB84. The situation is even worse for the class of continuous-variable qkd protocols where one party (Alice) only prepares coherent states from a (small) constellations of states. These discretely-modulated protocols are very promising for implementation, but so far there is no full security proof available in the literature that addresses the most general attacks in the finite-size setting.

The present paper fills this gap in the case of a protocol with a binary phase modulation (i.e. Alice sends either $|+a\rangle$ or $|-a\rangle$).

This is in my opinion an excellent piece of work and I will enthusiastically encourage publication in Nature Communications. However, as I will detail below, I have a few questions and suggestions that I hope the authors will address in a revised version of their manuscript.

Establishing the security of continuous-variable qkd protocols is notoriously difficult because the Hilbert space that describes Bob's system is an infinite-dimensional Fock space. When Alice's modulation is Gaussian, the protocols display a lot of symmetry which helps simplifying the problem. This is not the case anymore when Alice only sends coherent states from a discrete constellation. Before the present manuscript, to the best of my knowledge (and in agreement with what the authors state in their introduction), the only existing proofs were either assuming collective attacks in the asymptotic regime (i.e. forgetting about any finite-size effect) or looked at finite size effects for a restricted class of Gaussian attacks (which are probably not optimal).

The authors of the present manuscript go much further than the state-of-the-art by exploiting a clever mapping from the continuous-variable protocol to a qubit-based protocol. The basic idea here is to imagine a virtual protocol where Bob performs a QND measurement mapping his Fock space to a 2-dimensional space. While this idea is rather straightforward, the machinery developed by the authors to get a meaningful bound on the key rate of this virtual protocol is highly nontrivial and I find it quite impressive that they manage to get very good bounds on the key rate (despite the fact that we don't expect a binary modulation to allow for great performance in general).

It would be really great to extend this approach to larger constellations, as the authors discuss in the conclusion, but it is not obvious to me that this will be an easy task.

While I really like the results, I'd also like to share some comments and questions that I hope will help the authors to improve their manuscript even more.

Probably my main critique would be that the "security proof" section of the main text is very difficult to follow. I understand that proofs are always intricate and that conveying their intuition in a couple of pages is an impossible task, but I'd like to encourage the authors to give it a try nonetheless.

The security proof relies here on 4 different protocols:

- the real one (that is implemented in practice): Alice sends coherent states to Bob,
- a purified version: Alice prepares an initial entangled state, measures one subsystem and sends the other one to Bob (this is a standard step in most security proofs),
- a virtual protocol where Bob performs a QND map to his mode in order to get a qubit,
- a modified protocol allowing the authors to bound a useful quantity, related to the phase-error rate of typical qubit protocols.

I'm not exactly sure what would be the optimal way of introducing and discussing these four protocols: maybe a figure would be useful? In any case, I found that the present version of the

manuscript could be improved on this front.

This would also help reading the (long) method section where I found it sometimes hard to follow the various steps, mainly because I wouldn't always know which version of the protocol is discussed or how the different quantities/parameters relate to each other.

Because of that, I confess that I'm not completely sure to be able to have a clear picture in my head of how the general argument goes.

If I understand correctly, there is no space constraint for the method section, and it would therefore be great if the authors could spend some time better explaining how the proof works and how the different versions of the protocol compare to each other.

This would be tremendously useful for me, and I suspect also for many other potential readers.

I now continue with more specific comments/questions.

- Figure 1: maybe would it be useful to mention here that the values $m=1$, $r=0.4$ will be used for the security proof?

- on page 2, when introducing the proposed protocol, it would be useful to mention that $\beta = \sqrt{\mu}$. Maybe is it mentioned somewhere but couldn't find it.

- top of p.3, step 1 of the protocol is a bit mysterious because the reader doesn't understand what is the purpose of f_{suc} . It would be clearer to mention that this postselection is made to ensure that the error rate between Alice and Bob isn't too large.

- It would also be nice to discuss somewhere what is the value for the threshold of the step function f_{suc} . In the literature, as far as I know, the thresholds for this kind of postselection are always rather small in practice. Is it also the case here?

- step 4 of the protocol: why is it useful to encrypt the communication needed for error correction. Typical qkd protocols do not require this. I couldn't find any discussion of this point in the manuscript.

- step 5 of the protocol: what is s ?

- p.3 column 2: the bra $\langle x|$ isn't defined anywhere.

- p.4: it is not entirely clear how the virtual and the modified protocols differ. This is related to my general comment above that the proof strategy could be maybe explained more clearly.

- p.4: would it be possible to motivate a little bit the introduction of the variable T in eqn. (20). It appears very mysterious unless one really reads all the method section in detail.

- p.5: eqn.(28) is also very mysterious and the flow of the argument is a bit difficult to follow. It would be great if the authors could explain with more details in the "security proof section" what are the real difficulties to address to get a proof and defer the technical analysis to the "methods" section. The end of p.4 and beginning of p.5 are filled with technical steps but the reader (myself in any case) isn't quite sure about what we wish to prove.

For instance, the authors could explain somewhere that the purified version of the protocol is very

easy to define but in order to get a security proof similar to those for qubit-based protocols, we need to understand what happens when Alice performs an X-measurement instead of a Z-measurement. Since this is not something that can be implemented in practice, we need to come up with some tricks to analyse what would happen in this case. And this is the reason why the virtual protocol is introduced. All this is clear to the authors but not at all to the reader I think, and the paper would truly improve if it contained more explanations like this one.

At the moment, when reading the end of the security proof section, it is not clear why $T(\kappa, \gamma)$ is the right quantity to consider.

- p.5, in the numerical simulation section, it is not completely clear how the ξ value is defined. In particular, it is not obvious to me that the explanation in the text (multiplicative factor $(1+\xi)$) coincides with the explanation in figure 4 (additive noise of $\xi/2$).

- fig. 4: it would be great if you could say how large μ is in practice, as well as the value of x_{th} . I understand that these values are not constant (and you optimize over them), but still it would be nice and useful to see a rough estimate. This would also allow to compare the results here with those already present in the literature. In particular, how tight are the new bounds? do we lose a lot by addressing general attacks rather than just Gaussian attacks, etc.

- top of p.6: is it clear that (κ, γ) can be obtained by *convex* optimization?

- in the discussion p.6, the authors raise the question of a digitized outcome. This is a very good point. It is not clear to me what would happen in practice if the largest possible value ω is observed. Then one cannot expect to get any nonzero lower bound on $\langle 0|\rho|0\rangle$. Is this indeed what you also find because the function $\Lambda(m,r)$ is always negative for large r ?

- in general, the structure of the method section isn't very clear. Right now, it consists of many different subsection, but the general flow of the argument is missing. It would be great if another subsection could be added which details the general outline of the proof and then refers to the other subsections for specific lemmas.

Reviewer #2 (Remarks to the Author):

The authors of this manuscript describe a QKD protocol with continuous-variable systems (CV), with discrete modulation (binary) (DM), providing an original procedure/method based on a fidelity test that allows to “bound” a general eavesdropping and assess the security of the protocol against general attack. The methodology is inspired by the security proof of similar QKD protocol (BB92), originally designed for discrete-variable systems.

The work is interesting and timely, and the importance of the results relies on the fact that the fidelity-test method, described in the manuscript, allows somehow to “minimize” the assumption on the type of attacks implemented by the eavesdropper, then the security can be studied in a more general setup. Despite the security of the protocol is not proven in a composable scenario, the authors analyse the impact of finite-size effects (number of signals exchanged) on the key rate, that is promising for the next steps.

The work described in this manuscript should capture some interest from researchers working in the community of quantum cryptography, but it can be picked up also by researchers working in wider quantum information technology in general. The results are, in my opinion, correct and sound (see below for more detail).

Below some comment/point that authors may want to comment about

One of the motivation to be interested in CV-QKD protocols with discrete modulation is certainly the needs for digitalization conversion during signal processing, as mentioned by the authors. However, discrete-modulated CV-QKD are also expected to be extremely promising to allow long-distance security. The performance of the protocol in the manuscript, as described for example in figure 4(c), show a sharp decline already at around transmissivity 0.4 . This would represent few dB of attenuation and few km in fiber. That sounds a bit limited, also with respect to recent advances in Gaussian CV-QKD experiments, where hundreds of Km of distance range have been reported. The authors should clarify about this point.

Is the result depending on the specific security proof adopted?

Is the security bound tight? or it can be improved?

I would like the authors to comment regarding those points, and possible impact on research and implementation of DM CV-QKD.

The authors may want to consider the possibility to show the performance of their protocol (Figs. 4) plotting the gain against dB of attenuation, rather than transmissivity. I leave that to the authors' choice.

In the numerical analysis, the authors use an excess noise parameter ξ of about 10^{-3} . Some reference regarding this and other numbers should be clearly mentioned in the main text. It should be realistic, but I suggest they cite some up-to-date review and/or experimental paper in the literature.

I also ask the authors to comment about the fact that the inclusion of excess noise actually help to reduce $N=10^{12}$ to 10^{11} . One would expect that increasing the noise in the protocol, the possibility of error increase, i.e., p_{trash} should increase and with that also the number of signals needed to extract a secure key.

The methodology described in the main text is detailed in the Methods section.

Overall, the manuscript should be accessible to the specialist reader, however it could be less approachable for a non-expert audience.

I think that there is a review paper cited, but I think the bibliography of the manuscript should be completed, including recent reviews articles that may help the audience to frame the presented research within the general progress of quantum cryptography over the last years.

I would also be interested in seeing a comparison of the performance of the protocol against the linear bound giving the secret-key capacity. This can be helpful for understanding existing limitations and possible rooms for improvement. I suggest the authors include a new plot of the key-rate, possibly against dB or (better) distance (Km), for which the authors may use realistic state-of-the-art parameters.

The manuscript is generally sufficiently clear and well written.

For what previously said, I cannot recommend the present form of the manuscript for publication, but I am favourable to recommend it after the authors will amend and comment about my previous points.

Point-to-point response to Reviewers #1 and #2.

Reviewer #1

1. *Probably my main critique would be that the "security proof" section of the main text is very difficult to follow. I understand that proofs are always intricate and that conveying their intuition in a couple of pages is an impossible task, but I'd like to encourage the authors to give it a try nonetheless.*

We have made an extensive effort to revise the "security proof" section and the Methods along with the reviewer's suggestions 2,3,4,12,13, and 18 below, which we expect to help the readers to understand our results better.

2. *The security proof relies here on 4 different protocols:*

- *the real one (that is implemented in practice): Alice sends coherent states to Bob,*
- *a purified version: Alice prepares an initial entangled state, measures one subsystem and sends the other one to Bob (this is a standard step in most security proofs),*
- *a virtual protocol where Bob performs a QND map to his mode in order to get a qubit,*
- *a modified protocol allowing the authors to bound a useful quantity, related to the phase-error rate of typical qubit protocols.*

I'm not exactly sure what would be the optimal way of introducing and discussing these four protocols: maybe a figure would be useful? In any case, I found that the present version of the manuscript could be improved on this front.

After careful consideration, we decided that presenting three protocols is the best, where the 2nd and the 3rd protocols listed by the reviewer are merged to one. We gave distinct names to the three protocols and added a figure (Figure 4) to illustrate the relations between them.

3. *This would also help reading the (long) method section where I found it sometimes hard to follow the various steps, mainly because I wouldn't always know which version of the protocol is discussed or how the different quantities/parameters relate to each other.*

Because of that, I confess that I'm not completely sure to be able to have a clear picture in my head of how the general argument goes.

As stated above, we now explicitly present three distinct named protocols, which will help the readers to avoid confusion on which of the protocols are discussed in a context. In fact, the majority of the security proof is focused on deriving condition (16) in the estimation protocol, and there is no need to care about other protocols while proving (16). In the new manuscript, we have commented on this important fact (the security proof being reduced to proving an inequality in the estimation protocol) on three occasions: after Eq.(16) in the main text, at

the end of the caption for new figure 4, and in the Methods.

4. *If I understand correctly, there is no space constraint for the method section, and it would therefore be great if the authors could spend some time better explaining how the proof works and how the different versions of the protocol compare to each other.*

This would be tremendously useful for me, and I suspect also for many other potential readers.

In the old manuscript, the reason for the reduction to Eq. (16) was delegated to references. We have now added an extensive subsection in Methods (“2. Reduction to the estimation protocol”), which explains the rationale about the reduction in light of the relation between the three protocols.

5. *Figure 1: maybe would it be useful to mention here that the values $m=1$, $r=0.4$ will be used for the security proof?*

We added a comment on the adopted values to the caption of Figure 1.

6. *on page 2, when introducing the proposed protocol, it would be useful to mention that $\beta = \sqrt{\mu}$. Maybe is it mentioned somewhere but couldn't find it.*

After the description of the actual protocol, we added a comment “The parameter β is typically chosen to be $\sqrt{\eta \mu}$ with η being a nominal transmissivity of the quantum channel, while the security proof itself holds for any choice of β .”

7. *top of p.3, step 1 of the protocol is a bit mysterious because the reader doesn't understand what is the purpose of f_{suc} . It would be clearer to mention that this postselection is made to ensure that the error rate between Alice and Bob isn't too large.*

After the description of the actual protocol, we added a comment “The acceptance probability $f_{\text{suc}}(|x\rangle)$ should be chosen to post-select the rounds with larger values of $|x\rangle$, for which the bit error probability is expected to be lower. It is ideally a step function, but our security proof is applicable to any form of $f_{\text{suc}}(|x\rangle)$.”

8. *It would also be nice to discuss somewhere what is the value for the threshold of the step function f_{suc} . In the literature, as far as I know, the thresholds for this kind of postselection are always rather small in practice. Is it also the case here?*

We added tables in Methods to present examples of the optimized threshold value. The values of the threshold in our case are typically larger than those of post-selection protocols in the literature. We added a comment (p.6, right column), “Typical optimized values of the threshold x_{th} range from 0.4 to 1.5 (we adopted a normalization for which the vacuum

fluctuation is 0.5). They are larger than those in other analyses of protocols with post-selection (e.g., [30]). A possible reason is the fact that the latter protocols use more than two states to monitor the eavesdropping act, which may lead to a lower cost of privacy amplification and higher tolerance against bit errors.”

9. step 4 of the protocol: why is it useful to encrypt the communication needed for error correction. Typical qkd protocols do not require this. I couldn't find any discussion of this point in the manuscript.

The encryption is adopted in order to make the security analysis simpler, but it is possible to achieve the same net key rate without encryption. We added a paragraph explaining this at the end of the added subsection in Methods, “2. Reduction to the estimation protocol.”

10. step 5 of the protocol: what is s ?

After the description of the actual protocol, we added a comment “The parameters s and s' are related to the overall security parameter in the security proof below.”

11. p.3 column 2: the bra $\langle x|$ isn't defined anywhere.

We added a comment after Eq. (13) to explain that the bra maps a state vector to the value of its wave function at x .

12. p.4: it is not entirely clear how the virtual and the modified protocols differ. This is related to my general comment above that the proof strategy could be maybe explained more clearly.

As stated above, we sorted out relevant protocols and gave clear definition to each of them, which we believe help to avoid confusion.

13. p.4: would it be possible to motivate a little bit the introduction of the variable T in eqn. (20). It appears very mysterious unless one really reads all the method section in detail.

p.5: eqn.(28) is also very mysterious and the flow of the argument is a bit difficult to follow. It would be great if the authors could explain with more details in the "security proof section" what are the real difficulties to address to get a proof and defer the technical analysis to the "methods" section. The end of p.4 and beginning of p.5 are filled with technical steps but the reader (myself in any case) isn't quite sure about what we wish to prove.

For instance, the authors could explain somewhere that the purified version of the protocol is very easy to define but in order to get a security proof similar to those for qubit-based protocols, we need to understand what happens when Alice performs an X -measurement instead of a Z -measurement. Since this is not something that can be implemented in practice,

we need to come up with some tricks to analyse what would happen in this case. And this is the reason why the virtual protocol is introduced. All this is clear to the authors but not at all to the reader I think, and the paper would truly improve if it contained more explanations like this one.

At the moment, when reading the end of the security proof section, it is not clear why $T(\kappa, \gamma)$ is the right quantity to consider.

We are aware that Eq. (20) in the old manuscript may look mysterious for multiple reasons:

- i) It was not clear how it is related to the security of the actual protocol.
- ii) The linear form was presented all of a sudden with little motivation at this point.
- iii) If the behavior of the variable Q_- is independent of eavesdropping, why we bother to measure it in the estimation protocol?

As for i), the clarification of the protocols and the relation between them, which we carried out along with the reviewer's suggestions above, will be already enough for the readers to understand that we may focus on proving Eq. (16) in the estimation protocol, and there is no need to worry about the other protocols. As for ii), we reorganized and extended the flow of argument from (old) Eq. (20) to (28) in such a way that whenever the reader encounters a new expression, ample motivation should be given. We also explained that the computation of the function $B(\kappa, \gamma)$ is one of the difficulties we encountered. As for iii), we added an explanation about the reason in the paragraph including (new) Eq. (31), which is another difficulty we encountered and settled.

14. *p.5, in the numerical simulation section, it is not completely clear how the $\forall xi$ value is defined. In particular, it is not obvious to me that the explanation in the text (multiplicative factor $(1+\forall xi)$) coincides with the explanation in figure 4 (additive noise of $\forall xi/2$).*

For clarification, we revised the explanation in the text to "Bob receives Gaussian states obtained by randomly displacing coherent states $|+/- \sqrt{\eta \mu}\rangle$ to increase their variances" (p.6, right column).

The value $\forall xi/2$ in the previous manuscript was inconsistent with the quadrature scale chosen in the explanation of the model in Methods. The correct value should be $\forall xi/4$ since the vacuum state has a variance of $1/4$. We decided that a scale-independent description is better to avoid confusion and revised the figure caption to "the optical pulse that Bob receives is given by randomly displacing a coherent state to increase its variance by a factor of $(1+\forall xi)$."

15. *fig. 4: it would be great if you could say how large $\forall mu$ is in practice, as well as the value of x_{th} . I understand that these values are not constant (and you optimize over them), but still it would be nice and useful to see a rough estimate. This would also allow to compare the*

results here with those already present in the literature. In particular, how tight are the new bounds? do we lose a lot by addressing general attacks rather than just Gaussian attacks, etc.

We added tables in Methods to present examples of the optimized values of parameters. We also commented (p.7, right column) on comparison to recent asymptotic analyses [30, 31], and listed several possibilities for improvements in the proposed proof method. Nonetheless, we believe that the dominant reason why we cannot achieve the good key rates of [30, 31] is in the binary protocol we adopted. Hence the most promising route will be to extend our method to protocols with four or more states.

16. *top of p.6: is it clear that (κ, γ) can be obtained by *convex* optimization?*

We explained that the function $B(\kappa, \gamma)$ is convex (below Eq. (88)), and that U defined in Eq. (111) is also convex as a function of (κ, γ) .

17. *in the discussion p.6, the authors raise the question of a digitized outcome. This is a very good point. It is not clear to me what would happen in practice if the largest possible value $\forall\omega$ is observed. Then one cannot expect to get any nonzero lower bound on $\langle 0/\forall\rho/0 \rangle$. Is this indeed what you also find because the function $\Lambda(m,r)$ is always negative for large r ?*

In the case of an ideal heterodyne measurement with no bound on the outcome $\forall\omega$ (let us call it $\forall\omega_{\text{true}}$), a very strong pulse, which has a fidelity of almost zero, would produce a large value of $\forall\omega_{\text{true}}$. This is consistent with the fact that the function $\Lambda(\forall\nu)$ is always negative when $\forall\nu$ is large.

In the case of a realistic heterodyne measurement with a finite dynamic range, it will produce a saturated value $\forall\omega$ even when $|\forall\omega_{\text{true}}|$ is much larger than $|\forall\omega|$. If $|\forall\omega|$ is sufficiently large, the strategy proposed in Discussion section dictates that we should use $\Lambda(|\forall\omega - \forall\beta|^2)$ to maintain security. This is because the slope of $\Lambda(\forall\nu)$ for a large value of $\forall\nu$ is always positive, and hence $\Lambda(|\forall\omega - \forall\beta|^2) = \langle \Lambda(|\forall\omega_{\text{true}} - \forall\beta|^2) \rangle$ holds.

18. *in general, the structure of the method section isn't very clear. Right now, it consists of many different subsection, but the general flow of the argument is missing. It would be great if another subsection could be added which details the general outline of the proof and then refers to the other subsections for specific lemmas.*

We made an effort to improve the structure of the Methods. We reorganized the parts describing the security proof into four numbered subsections, headed by a paragraph explaining the roles and relations of the four subsections. We also presented several key

statements as propositions (corollaries), which will help to clarify the logical dependency among the subsections.

Reviewer #2

1. *Despite the security of the protocol is not proven in a composable scenario, the authors analyse the impact of finite-size effects (number of signals exchanged) on the key rate, that is promising for the next steps.*

We would like to point out that our security proof adopted the security condition that is standard in QKD (DV and CV alike) and is known to satisfy the universal composability. In the manuscript, it is stated at the end of subsection “1. Definition of security in the finite size regime” in Methods (p.9, left column).

2. *One of the motivation to be interested in CV-QKD protocols with discrete modulation is certainly the needs for digitalization conversion during signal processing, as mentioned by the authors. However, discrete-modulated CV-QKD are also expected to be extremely promising to allow long-distance security. The performance of the protocol in the manuscript, as described for example in figure 4(c), show a sharp decline already at around transmissivity 0.4. This would represent few dB of attenuation and few km in fiber. That sounds a bit limited, also with respect to recent advances in Gaussian CV-QKD experiments, where hundreds of Km of distance range have been reported.*

The authors should clarify about this point.

Is the result depending on the specific security proof adopted?

Is the security bound tight? or it can be improved?

I would like the authors to comment regarding those points, and possible impact on research and implementation of DM CV-QKD.

We commented (p.7, right column) on several possibilities for improvements in the proposed proof method. Then we stated our belief that the dominant reason why we cannot achieve long distance security is in the binary protocol we adopted. We thus expect that by extending our method (or part thereof) to protocols with four or more states, there is a good chance that we can enjoy long distance security with a high level of security. We believe that the impact of our result lies in opening up this new direction.

3. *The authors may want to consider the possibility to show the performance of their protocol (Figs. 4) plotting the gain against dB of attenuation, rather than transmissivity. I leave that to the authors' choice.*

We adopted log-log plots for Figure 4 (new Figure 5) as suggested by the reviewer.

4. *In the numerical analysis, the authors use an excess noise parameter ξ of about 10^{-3} . Some reference regarding this and other numbers should be clearly mentioned in the main text. It should be realistic, but I suggest they cite some up-to-date review and/or experimental paper in the literature.*

We cited (p.6, right column) experiments [7, 45, 48] which suggest that excess noise with $\xi=10^{-2} - 10^{-3}$ at the channel output seems reasonable. The only other parameter in our model that is related to technology level is the total number N of pulses. For this, we cited the experiment in Ref. [7] with a repetition rate of 0.5GHz and demonstration of wavelength division multiplexing to increase the effective rate further, which suggests that $N=10^{12}$ is well within the current technology.

5. *I also ask the authors to comment about the fact that the inclusion of excess noise actually help to reduce $N=10^{12}$ to 10^{11} . One would expect that increasing the noise in the protocol, the possibility of error increase, i.e., p_{trash} should increase and with that also the number of signals needed to extract a secure key.*

We added a comment (p6, right column) on a possible reason behind this behavior, “This may be ascribed to the cost of the fidelity test. In order to make sure that the fidelity is no smaller than $1-\delta$, the statistical uncertainty of the fidelity must be reduced to $O(\delta)$. As a result, approaching the asymptotic rate of $\xi=0$ will require many rounds for the fidelity test.”

6. *The methodology described in the main text is detailed in the Methods section.*

Overall, the manuscript should be accessible to the specialist reader, however it could be less approachable for a non-expert audience.

I think that there is a review paper cited, but I think the bibliography of the manuscript should be completed, including recent reviews articles that may help the audience to frame the presented research within the general progress of quantum cryptography over the last years.

Very recently a comprehensive review on practical QKD covering DV- and CV-QKD has been published. We cited this (Ref. [6]) for the readers to be able to learn state-of-art research progress on QKD.

7. *I would also be interested in seeing a comparison of the performance of the protocol against the linear bound giving the secret-key capacity. This can be helpful for understanding existing limitations and possible rooms for improvement. I suggest the authors include a new plot of the key-rate, possibly against dB or (better) distance (Km), for which the authors may use*

realistic state-of-the-art parameters.

We added the curve for the repeaterless bound to the plot with $\forall \xi=0$ (new Figure 5c) as suggested by the reviewer. As we mentioned above, we believe that the values of N and ξ used in the plots in new Figure 5a)-d) are realistic.

Changes made for improving readability:

Here we list the changes that do not correspond to specific comments of the reviewers but were made to improve the presentation of the manuscript.

We changed the variable $\forall \mu$ used for the argument of functions and for integrations to $\forall \nu$ in p.2 (left column), Fig. 1a), and p.8 (left column) to avoid confusion with the protocol parameter $\forall \mu$ for the intensity of the optical pulse.

For clarification, we have assigned distinct names for the optical pulse emitted from Alice and for that received by Bob. Both were named 'B' in the old manuscript but are now ' $\forall \tilde{C}$ ' and 'C'. This notation is also included in Figure 2. Bob's qubit is now called plain B instead of B' in the old manuscript.

We changed the operator $\forall \Pi_-$ defined in (old) Eq. (23) to $\forall \Pi^{\{\text{trash}\}}_-$ to avoid confusion with $\forall \Pi_-$ used in Lemma 1 in the Methods. Definition of $\forall \Pi_+$ in the same equation was omitted since we no longer use it.

Changes made for figures:

Figure 1 a):

The value of parameter r for each of the three curves are now chosen such that it minimizes the range of the Λ function.

Figure 2:

The names ' $\forall \tilde{C}$ ' and 'C' of the optical pulses used in the text were added in the diagram for convenience.

Figure 3:

b' =even and b' =odd were changed to b' =+ and b' =- for the consistency with the revision of the main text.

Figure 4: It was newly added.

Figure 5 (old Fig. 4):

The abscissa of each graph was changed to log-scale (in dB). The curve for the PLOB bound was added to c).

Reviewer #1 (Remarks to the Author):

I'm satisfied with the answers provided by the authors, as well as the new version of the manuscript. I'm favorable to the publication of this manuscript in Nature Communications.

Reviewer #2 (Remarks to the Author):

The authors have amended my points and overall my opinion on the manuscript is positive. I ask the authors to update the bibliography including the following review paper on quantum cryptography, arXiv:1906.01645

It should be added to the bibliography, more or less where now Ref. [6] is.

Also, the main text should be modified, somewhere, in the first paragraph of the Introduction, to include something along the following line

"[...] Ref. [arXiv:1906.01645] provides an extensive review of recent progress in quantum cryptography."

After the authors have implemented these changes, in the bibliography and in the main text, I think the manuscript can be recommend for publication.